# Principled SVD-based Delta Compression via Quantization Error Minimization

Boya Xiong [1]   Shuo Wang [2]   Weifeng Ge [3]   Guanhua Chen [4]   Yun Chen [1 5]

## Abstract

Supervised Fine-Tuning (SFT) empowers Large Language Models (LLMs) with exceptional performance on specialized tasks, but it yields dense, high-dimensional delta parameters that pose severe storage and distribution challenges. Singular Value Decomposition (SVD)-based compression offers a compact representation for such delta parameters, but existing methods adopt heuristic quantization without clarifying underlying mechanisms, leading to poor generalizability. In this work, we propose PRINMIX, a rigorous SVD-based framework that models quantization as an optimization problem, grounding the design in mathematical mechanisms. We first theoretically derive quantization error and identify a key singular-value-dominated scaling mechanism, which provides a formal motivation for mixed-precision quantization under the GPTQ-style reconstruction objective. We then model the quantization scheme as a 0/1 Integer Linear Programming (ILP) problem, which yields optimal bit-budget-constrained solutions without empirical assumptions. Furthermore, PRINMIX integrates a Reconstruction Target Correction (RTC) method to compensate for errors from the $\mathbf{V}$-then-$\mathbf{U}$ sequential quantization process. Extensive experiments confirm PRINMIX performs well: for 7B LLMs, PRINMIX outperforms SOTA Delta-CoMe on challenging benchmarks by 22.3% on AIME2024 and 6.1% on GQA.

## 1. Introduction

Large language models (LLMs) have shown breakthrough performance on various knowledge-intensive (Grattafiori et al., 2024; Team, 2024; Zhu et al., 2026; Zhang et al., 2026b), evaluation (Zhang et al., 2026a;d;c; Deng et al., 2025), and complex reasoning tasks (DeepSeek-AI, 2025; Li et al., 2026; Huang et al., 2026; Zhu et al., 2025). Enhancing deployment efficiency is crucial for facilitating LLM applications on edge devices and in cloud environments (Yao et al., 2024). In multi-tenant serving scenarios, multiple users fine-tune the same base model using their customized datasets (Wei et al., 2024; Yu et al., 2023), resulting in a variety of customized models that share a common foundation. These models, derived from the same base LLM (e.g., Qwen2.5 (Team, 2024) or LLaMA (Grattafiori et al., 2024)), need to be deployed concurrently to address simultaneous user requests. Conventional LLM compression approaches (Frantar et al., 2022; Lin et al., 2024) focus on quantizing and compressing the full model parameters. While effective at low compression ratios, these methods struggle to maintain model performance at high compression ratios, resulting in significant storage and computational overhead when deploying multiple customized LLMs.

In contrast to full model compression, delta compression (Yao et al., 2024; Liu et al., 2024; Ping et al., 2024) decomposes a customized LLM into two components: the base model and the delta weights, which encapsulate the differences between the customized model and its corresponding base model. This approach emphasizes the compression of delta weights. Consequently, in multi-tenant environments, a single base model can be deployed alongside multiple sets of compressed delta parameters. Delta compression achieves significantly higher compression rates than full model compression, thereby substantially reducing overall deployment costs. Researchers have explored effective approaches for delta compression. Ryu et al. (2023) leverages the low-rank characteristics of delta weights to improve storage efficiency through low-rank approximation. Liu et al. (2024) proposes a 1-bit quantization approach, termed BitDelta, to reduce the size of delta weights. Delta-CoMe (Ping et al., 2024) introduces a mixed-precision delta compression technique based on singular value decomposition (SVD), allocating higher-bit representations to singular vectors associated with larger singular values. Although these existing approaches demonstrate promising performance at high compression ratios, they lack rigorous mathematical foundations, which can lead to suboptimal performance,

---

[1]Shanghai University of Finance and Economics [2]Tsinghua University [3]Fudan University [4]Southern University of Science and Technology [5]MoE Key Laboratory of Interdisciplinary Research of Computation and Economics. Correspondence to: Guanhua Chen <chengh3@sustech.edu.cn>, Yun Chen <yunchen@sufe.edu.cn>.

*Proceedings of the 43rd International Conference on Machine Learning*, Seoul, South Korea. PMLR 306, 2026. Copyright 2026 by the author(s).

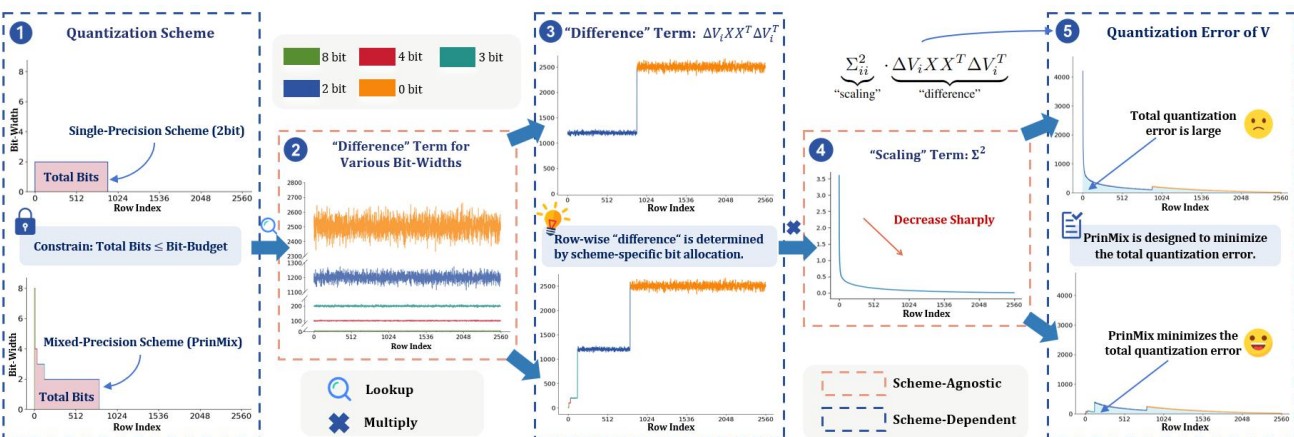

*Figure 1.* An overview of PRINMIX compared to single-precision quantization. Given the quantization scheme (①), we compute the "difference" term (③) by looking up the corresponding values in subfigure ②. The quantization error of the $i$-th row of $\mathbf{V}$ comprise two components: a "scaling" term (④) and a "difference" term (③). PRINMIX identifies the optimal quantization scheme within the constraints of the bit budget (①) to effectively balance these two components, thereby minimizing the total quantization error of $\mathbf{V}$ (⑤). Note that the "difference" term for various bit-widths (②) is pre-computed using a calibration dataset and remains fixed during the optimization process.

especially in challenging compression scenarios.

In this work, we propose PRINMIX, a high-performance principled mixed-precision delta compression framework grounded in a solid theoretical foundation. PRINMIX implements delta compression within the SVD space, formulating the quantization objective as the minimization of layer-wise quantization error. By pursuing this objective, PRINMIX establishes a mathematically sound mixed-precision compression strategy that accommodates flexible, user-defined compression ratios. This strategy derives **the mixed-precision scheme** through the solution of a 0/1 linear integer programming problem and ensures optimization consistency throughout the quantization process via **a reconstruction target correction method**. Unlike Ping et al. (2024), which empirically posits that singular vectors corresponding to larger singular values are more significant and, therefore, necessitate higher-bit representations, PRINMIX prioritizes the minimization of quantization error. It formulates all subsequent strategies based exclusively on this principle, eschewing reliance on singular values for assessing importance. This distinction is vital, as prior research has demonstrated that the significance attributed to singular values may not correlate with the performance of LLMs (Hsu et al., 2022; Wang et al., 2025).

We conduct extensive experiments on reasoning, math, code, and multimodal tasks across eight aligned LLMs to demonstrate the effectiveness of PRINMIX. The results show that PRINMIX achieves state-of-the-art performance among delta compression methods, particularly in challenging scenarios where the norm of $\Delta\mathbf{W}$ is large. Notably, on the reasoning task AIME2024, PRINMIX surpasses the leading baseline, Delta-CoMe, by 22.3% on the 7B model and 26.9% on the 14B model. Furthermore, PRINMIX can

achieve more than 6× GPU memory and disk storage savings, enabling the deployment of multiple models within constrained resource environments.

## 2. Related Work

**Quantization Strategies for LLMs** Quantization reduces the bit-precision of model parameters to lower GPU cost and accelerate inference. Current strategies for LLM quantization can be broadly categorized into quantization-aware training (QAT) and post-training quantization (PTQ). QAT simulates quantization operations during training and uses backpropagation to correct quantization errors (Zhou et al., 2018; Esser et al., 2020; Liu et al., 2023b; Wang et al., 2023). In contrast, PTQ quantizes a pre-trained model without further training, typically calibrating the quantized weights with a modest calibration dataset (Dettmers et al., 2022; Frantar et al., 2022; Lin et al., 2024; Lee et al., 2024). Given the high computational cost associated with training or fine-tuning large language models, PTQ has become a particularly prevalent approach for LLM quantization. In our work, we leverage the GPTQ (Frantar et al., 2022) method within PTQ, focusing on mixed-precision quantization of the singular vectors of the delta parameters.

**Delta Compression** Delta compression (Isik et al., 2023; Ryu et al., 2023; Liu et al., 2024; Ping et al., 2024) aims to diminish the storage and inference costs associated with serving multiple models by compressing delta parameters, which are the differences between the parameters of a fine-tuned LLM and its corresponding base LLM. GPT-Zip (Isik et al., 2023) extends GPTQ to compress the delta parameters into 2-bit, and then sparsify 95% of the quantized delta weights to further reduce storage costs. DeltaZip (Yao

**Algorithm 1** Algorithm for Quantization in PRINMIX

**Data:** Delta parameter $\mathbf{W}$, List of candidate quantization bits $Q$, predefined averaged bit-width $G_b$, Calibration set $X$

**Result:** Quantized matrices $\hat{\mathbf{V}}$ and $\hat{\mathbf{U}}$

$\mathbf{U}, \mathbf{\Sigma}, \mathbf{V} \leftarrow \text{SVD}(\mathbf{W})$

**for** *bit b in Q* **do**

    $\mathbf{V_b} \leftarrow \text{SimQuant}(\mathbf{V}, b, X)$

    $\mathbb{E}_b^V \leftarrow \text{CalcLoss}(\mathbf{V}, \mathbf{V_b}, \mathbf{\Sigma})$

**end**

$B \leftarrow \text{CalcStorage}(Q)$

$S \leftarrow \text{SolveOpt}(B, G_b, \mathbb{E}^{\mathbb{V}})$

$\hat{\mathbf{V}} \leftarrow \text{QuantParams}(\mathbf{V}, \mathbf{S}, X)$

$\tilde{\mathbf{U}} \leftarrow \text{RTC}(\mathbf{U}, \hat{\mathbf{V}}, \mathbf{V}, \mathbf{\Sigma}, X)$

$\hat{\mathbf{U}} \leftarrow \text{QuantParams}(\tilde{\mathbf{U}}, \mathbf{S}, \hat{\mathbf{V}}, \mathbf{\Sigma}, X)$

**return** $\hat{\mathbf{V}}, \hat{\mathbf{U}}$;      // Return results

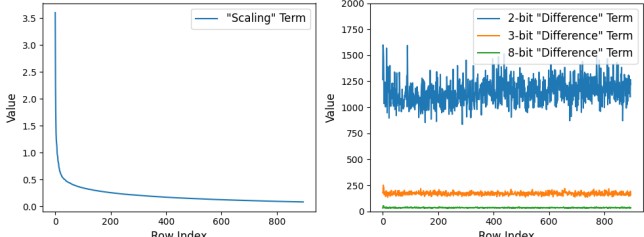

*Figure 2.* (Left) The value of "scaling" term (Eq. 5) at different row indices. (Right) The value of "difference" term (Eq. 5) with different quantization bit-width at different row indices. We compute all results using Q_Proj at the last layer of Qwen2.5-Math-7B-Instruct.

et al., 2024) extends the idea of structured pruning and delta compression to develop a multi-tenant serving system. However, both methods are still limited to compression ratios of 2-bit and higher. Liu et al. (2024) introduces BitDelta, which compresses delta weight into 1-bit, using a trainable high-precision scaling factor for each delta weight matrix. From this point onward, the compression of delta parameters has entered the 1-bit era. In addition to these low-bit methods, Ryu et al. (2023) identifies the low-rank property of delta weights and achieves delta compression through low-rank approximation. Recently, Delta-CoMe (Ping et al., 2024) leverages the benefits of both low-rank and low-bit compression methods, proposing a mixed-precision delta compression method that uses varying bit-widths to represent different singular vectors of the delta weights. However, the rationale behind their mixed-precision quantization is predicated on a questionable hypothesis (Hsu et al., 2022; Wang et al., 2025): that singular vectors associated with larger singular values are inherently more important. This premise lacks a solid theoretical foundation, leading to a mixed-precision strategy that is primarily empirical and, consequently, suboptimal. In this work, we introduce PRIN-MIX, which theoretically explains why mixed-precision is favored under the GPTQ-style layer-wise reconstruction objective and a global bit-budget constraint, and derives a quantization approach based on our theoretical analysis.

## 3. Method

In this section, we introduce PRINMIX, an adaptive mixed-precision delta compression strategy for LLMs with mathematical support. In Section 3.1, we begin with the minimization of quantization error in the SVD space and derive the detailed $\mathbf{V}$-then-$\mathbf{U}$ sequential quantization process. We provide a formal analysis explaining when mixed-precision is favored in this context. In Section 3.2, we present the detailed design of our mixed-precision scheme, which is formulated as a 0/1 Integer Linear Programming (ILP) problem.

Algorithm 1 shows the details of PRINMIX.

### 3.1. Quantization Error Derivation

At a high level, PRINMIX follows the structure of the classical post-training quantization method GPTQ, by performing quantization to minimize the reconstruction error. Given a delta weight matrix $\mathbf{W}$ and the corresponding input $X$, the quantization objective of the GPTQ is to find a quantized matrix $\hat{\mathbf{W}}$ which minimizes the squared error:

$$\arg\min_{\hat{\mathbf{W}}} \left\| \mathbf{W}X - \hat{\mathbf{W}}X \right\|_F^2 = \sum_i \left\| W_i X - \hat{W}_i X \right\|_F^2 = \sum_i e_i \tag{1}$$

Following previous work (Hassibi et al., 1993; Nagel et al., 2020), the quantization error of the $i^{\text{th}}$ row of $\mathbf{W}$ can be exactly expressed as the following quadratic form $e_i$:

$$e_i = \frac{1}{2}\Delta W_i \mathbf{H}_i \Delta W_i^T \tag{2}$$

Here $\Delta W_i = W_i - \hat{W}_i$ is the quantization difference of $i^{\text{th}}$ row, while the Hessian matrix $\mathbf{H}_i = 2XX^T$ is independent and identical across different rows in $\mathbf{W}$. By reusing $\mathbf{H}$, GPTQ derives the optimal quantized weights $\hat{\mathbf{W}}$ row by row, allowing for parallel computation across multiple rows.

Instead of directly quantizing $\mathbf{W}$, PRINMIX performs quantization in the SVD space, by finding a quantized matrix $\hat{\mathbf{U}}$ and $\hat{\mathbf{V}}$ which minimizes the squared error:

$$\arg\min_{\hat{\mathbf{U}}, \hat{\mathbf{V}}} \left\| \mathbf{U}\mathbf{\Sigma}\mathbf{V}X - \hat{\mathbf{U}}\mathbf{\Sigma}\hat{\mathbf{V}}X \right\|_F^2 \tag{3}$$

where $\mathbf{W} = \mathbf{U}\mathbf{\Sigma}\mathbf{V}$. Below, we introduce the detailed $\mathbf{V}$-then-$\mathbf{U}$ sequential quantization process of PRINMIX, which first quantizes $\mathbf{V}$, and then moves to $\mathbf{U}$.

#### 3.1.1. QUANTIZE $\mathbf{V}$

In this section, we present a theoretical analysis that motivates the need for mixed-precision quantization. Specif-

ically, we find the quantized $\hat{\mathbf{V}}$ with the row-by-row approach by minimizing the squared error:

$$\arg\min_{\hat{\mathbf{V}}} \left\| \mathbf{U}\boldsymbol{\Sigma}\mathbf{V}X - \mathbf{U}\boldsymbol{\Sigma}\hat{\mathbf{V}}X \right\|_F^2 = \sum_i e_i^{\mathbf{V}}$$

$$e_i^{\mathbf{V}} = \frac{1}{2}\Delta V_i \mathbf{H}_i^{\mathbf{V}} \Delta V_i^T \tag{4}$$

Here $\Delta V_i = V_i - \hat{V}_i$ is the quantization difference of the $i^{\text{th}}$ row, and $\mathbf{H}_i^{\mathbf{V}} = 2\Sigma_{ii}^2 \cdot XX^T$ is the Hessian matrix of the $i^{\text{th}}$ row of $\mathbf{V}$ (with derivation details in Appendix A.1). As $\Sigma_{ii}^2$ is a scalar, we can reformulate the Eq. (4) as follows:

$$e_i^{\mathbf{V}} = \frac{1}{2}\Delta V_i \mathbf{H}_i^{\mathbf{V}} \Delta V_i^T = \underbrace{\Sigma_{ii}^2}_{\text{"scaling"}} \cdot \underbrace{\Delta V_i XX^T \Delta V_i^T}_{\text{"difference"}} \tag{5}$$

From Eq. (5), it is evident that the error for $i$-th row of $\mathbf{V}$ comprises two components: a "scaling" term $\Sigma_{ii}^2$, which suggests that rows (singular vectors) with larger singular values has larger scaling factor, and a "difference" term $\Delta V_i XX^T \Delta V_i^T$, derived from the quantization differences $\Delta V_i$ and limited sampling over a calibration set.

As illustrated in Figure 2, we present the results of the "scaling" and "difference" terms across different rows. The variation in the "difference" term remains relatively minor when the same bit-width is used to quantize different rows. In contrast, the "scaling" term decreases sharply as the row index $i$ increases. Consequently, the quantization error $e_i^{\mathbf{V}}$, which encompasses both terms, varies significantly across different rows under a uniform bit-width for quantization. To minimize the total error, it is ideal for the quantization error of each row to be small. Given that the "scaling" term is fixed for each row, we can only adjust the "difference" term by carefully allocating bit-widths. However, due to the constraints of the total bit budget, we cannot allocate high bit-widths to all rows simultaneously. Therefore, we propose a strategy of assigning varying bit-widths to different rows to reduce the overall quantization error.

### 3.1.2. QUANTIZE $\mathbf{U}$

In this section, we analyze why mixed-precision quantization is not crucial for $\mathbf{U}$. After quantizing $\mathbf{V}$ to $\hat{\mathbf{V}}$, the quantization objective of $\mathbf{U}$ is:

$$\arg\min_{\hat{\mathbf{U}}} \|\mathbf{U}\boldsymbol{\Sigma}\hat{\mathbf{V}}X - \hat{\mathbf{U}}\boldsymbol{\Sigma}\hat{\mathbf{V}}X\|_F^2 = \sum_i e_i^{\mathbf{U}}$$

$$e_i^{\mathbf{U}} = \frac{1}{2}\Delta U_i \mathbf{H}_i^{\mathbf{U}} \Delta U_i^T = \Delta U_i \boldsymbol{\Sigma}\hat{\mathbf{V}}XX^T\hat{\mathbf{V}}^{\mathbf{T}}\boldsymbol{\Sigma}^{\mathbf{T}} \Delta U_i^T \tag{6}$$

Here $\Delta U_i = U_i - \hat{U}_i$, and the Hessian matrix of the $i^{\text{th}}$ row of $\mathbf{U}$ is given by $\mathbf{H}_i^{\mathbf{U}} = 2\boldsymbol{\Sigma}\hat{\mathbf{V}}XX^T\hat{\mathbf{V}}^{\mathbf{T}}\boldsymbol{\Sigma}^{\mathbf{T}}$ (with derivation details in Appendix A.2). Upon comparing Eq. (5) and Eq.

(6), we observe that $e_i^{\mathbf{U}}$ does not incorporate the scaling term present in Eq. (5). Consequently, when different rows are quantized using the same bit-width, there is no significant variation in error. This uniformity arises from the fact that the Hessian matrices for different rows of $\mathbf{U}$ are identical. Thus, unlike $\mathbf{V}$, there is no necessity to employ mixed-precision when quantizing different rows of $\mathbf{U}$.

Therefore, PRINMIX **determines the mixed-precision quantization schedule based on $\mathbf{V}$**, and then applies the same schedule to $\mathbf{U}$ for simplicity. Specifically, PRINMIX quantizes $\mathbf{U}$ using a column-wise mixed-precision schedule, where the $i^{\text{th}}$ column of $\mathbf{U}$ adopts the same bit-width as the $i^{\text{th}}$ row of $\mathbf{V}$ as they correspond to the same singular value. Notably, PRINMIX exhibits insensitivity to column-wise precision schedules, since GPTQ compensates for quantization-induced errors in the column direction by adjusting the unquantized weights during the quantization process. This compensation, however, does not occur between different rows, as different rows are independently quantized in GPTQ. This further underscores the importance of discussing row-wise mixed-precision strategies aimed at minimizing the quantization error of $\mathbf{V}$. To empirically verify whether $\mathbf{U}$ is sensitive to the quantization schedule, we compare different quantization schemes for $\mathbf{U}$ in Appendix C.1. The results show that applying the same schedule used for $\mathbf{V}$'s rows to $\mathbf{U}$'s columns yields performance comparable to applying uniform and other mixed-precision schemes to $\mathbf{U}$ . This further supports our claim that $\mathbf{U}$ is insensitive to the quantization schedule.

**Reconstruction Target Correction** In Eq. (6), we quantize $\mathbf{U}$ to reconstruct the target $\mathbf{U}\boldsymbol{\Sigma}\hat{\mathbf{V}}X$, which deviates from the initial target $\mathbf{U}\boldsymbol{\Sigma}\mathbf{V}X$. This deviation can negatively impact the performance of the quantized model. A straightforward approach to address this issue is to directly replace the reconstruction target with $\mathbf{U}\boldsymbol{\Sigma}\mathbf{V}X$; however, this would inhibit the application of GPTQ for quantization, as it breaks the premise that GPTQ relies solely on second-order terms. Therefore, we propose a method termed "Reconstruction Target Correction" (RTC) to reduce the bias by transforming $\mathbf{U}\boldsymbol{\Sigma}\hat{\mathbf{V}}X$ in Eq. (6) to $\tilde{\mathbf{U}}\boldsymbol{\Sigma}\hat{\mathbf{V}}X$, where $\tilde{\mathbf{U}}$ is derived from the following equation:

$$\min_{\tilde{\mathbf{U}}} \left\| \mathbf{U}\boldsymbol{\Sigma}\mathbf{V}X - \tilde{\mathbf{U}}\boldsymbol{\Sigma}\hat{\mathbf{V}}X \right\|_F^2$$

$$\Rightarrow \tilde{\mathbf{U}} = \mathbf{U}\boldsymbol{\Sigma}\mathbf{V}XX^T\hat{\mathbf{V}}^{\mathbf{T}}\boldsymbol{\Sigma}^{\mathbf{T}}(\boldsymbol{\Sigma}\hat{\mathbf{V}}XX^T\hat{\mathbf{V}}^{\mathbf{T}}\boldsymbol{\Sigma}^{\mathbf{T}})^{-1} \tag{7}$$

See Appendix A.3 for detailed derivations. We further discuss the numerical stability of the matrix inverse in Eq. (7), including Cholesky decomposition with diagonal regularization and per-layer condition number statistics, in Appendix C.4. In summary, prior to quantizing $\mathbf{U}$, we first update $\mathbf{U}$ to $\tilde{\mathbf{U}}$ using Eq. (7). Subsequently, we perform quantization by minimizing $\|\tilde{\mathbf{U}}\boldsymbol{\Sigma}\hat{\mathbf{V}}X - \hat{\mathbf{U}}\boldsymbol{\Sigma}\hat{\mathbf{V}}X\|_F^2$. This approach

*Table 1.* Selected backbone and aligned models for the examined four tasks.

| Task | 7B Models | | 13-14B Models | |
|---|---|---|---|---|
| | Backbone | Aligned | Backbone | Aligned |
| Math | Qwen2.5-Math | Qwen2.5-Math-Instruct | LLaMA2 | MetaMath |
| Reasoning | Qwen2.5-Math | DeepSeek-R1-Distill-Qwen | Qwen2.5 | DeepSeek-R1-Distill-Qwen |
| Coder | Qwen2.5-Coder | Qwen2.5-Coder-Instruct | Qwen2.5-Coder | Qwen2.5-Coder-Instruct |
| Multimodal | Qwen2.5 | Qwen2.5-VL-Instruct | LLaMA2 | LLAVA-V1.5 |

aims to ensure that the reconstruction target closely approximates the original, without compromising the application of GPTQ for quantization.

### 3.2. Optimization Problem Modeling

In this section, we formulate the optimal mixed-precision bit allocation problem as a 0/1 integer linear programming model (see Eq. (8)). Given a user-specified compression target bit $G_b$, a candidate set of quantization bit-widths $Q$ of size $N_b$, and an upper bound $f_{max}$ on the number of active bit-widths, the proposed model minimizes the quantization error by automatically selecting a subset of active bit-widths from $Q$, subject to the constraints imposed by $G_b$ and $f_{max}$.

$$\min_{\mathcal{S}} \sum_i \mathbb{E}_i^{\mathbf{V}} \mathcal{S}_i^T \quad \text{(Total quantization error)}$$

$$\text{s.t.} \sum_i \mathcal{S}_i B \leq G_b(h_{in} \cdot h_{out}) \quad \text{(Bit budget constraint)}$$

$$\text{sum}(S_i) = 1 \quad \text{(One-hot vector constraint)}$$

$$S_i - f \leq 0 \quad \text{(Bit-width selection constraint)}$$

$$\text{sum}(f) \leq f_{max} \quad \text{(Bit-width number constraint)}$$

$$(8)$$

As shown in Eq. (8), the objective is to minimize the total quantization error, expressed as $\sum_i \mathbb{E}_i^{\mathbf{V}} \mathcal{S}_i^T$. Here, $\mathbb{E}_i^{\mathbf{V}} \in \mathbb{R}^{1 \times N_b}$ denotes the quantization error associated with different bit-widths for the $i^{th}$ row of $\mathbf{V}$, computed using predefined calibration data samples $X_n$ in accordance with Eq. (4). $\mathcal{S}_i \in \mathbb{R}^{1 \times N_b}$ is a binary optimization variable indicating the selected bit-width for quantizing the $i^{th}$ row of $\mathbf{V}$ and the corresponding $i^{th}$ column of $\tilde{\mathbf{U}}$. Note that our objective is limited to the quantization error of $\mathbf{V}$, with a detailed discussion provided in Sections 3.1.1 and 3.1.2.

The optimization problem has four constraints. (1) The "bit-budget constraint" ensures that the quantized model achieves a target compression bit that does not exceed the predefined threshold $G_b$. Here $h_{in}$ and $h_{out}$ represent the input and output dimension of $\mathbf{W}$. $B \in \mathbb{R}^{N_b \times 1}$ represents the storage required for quantizing a row of $\mathbf{V}$ and a column of $\tilde{\mathbf{U}}$ at different bit-widths, which is computed as $B = (h_{in} + h_{out}) \cdot Q$. (2) The "one-hot vector constraint" requires that each row of $\mathbf{V}$ and the corresponding column of $\tilde{\mathbf{U}}$ be quantized using exactly one bit-width. (3) The "bit-width selection constraint" guarantees that only permis-

sible bit-widths are utilized for quantization. The variable $f \in \mathbb{R}^{1 \times N_b}$ denotes the set of admissible bit-widths, where $f_{0,k} = 1$ indicates that the $k^{th}$ bit-width in $Q$ is allowable. (4) The "bit-width number constraint" restricts the number of admissible bit-widths to a maximum of $f_{max}$.

The 0/1 integer linear programming optimization problem is then solved with the CVXPY (Diamond & Boyd, 2016) library and the SCIP (Maher et al., 2016) solver. For the 7B model, the SCIP takes 0.5 hours to solve the ILP problem and takes 3.9 hours scaling to the 70B model. The solving time can be further reduced by at least $4\times$ by switching to a commercial solver (Ge et al., 2023) and reducing the solution space. We also introduce dynamic programming (DP) as an alternative solver to minimize the same quantization error defined in Eq. (8). Compared with the ILP model, DP is computationally efficient; however, it cannot restrict the number of distinct bit-widths used in the mixed-precision schedule. We discuss the acceleration methods and DP solver in detail in Appendix C.5.

By solving Eq. (8), we obtain an optimal mixed-precision scheme that minimizes the error while satisfying predefined bit budget constraints. This allows us to derive task-specific mixed-precision quantization strategies which balance the "scaling" and "difference" terms, leading to improved performance across various tasks.

## 4. Experiments

### 4.1. Experiment Setup

**Evaluation Tasks** We evaluate our methods on four distinct tasks: reasoning, math, code generation, and multimodal. These tasks encompass a vast array of current directions based on fine-tuning with open-source LLMs. **Reasoning:** We use the Math500 and AIME2024 datasets as the test set. **Math:** We use the GSM8K (Cobbe et al., 2021) and Math500 (Lightman et al., 2023) datasets as the test set. **Code Generation:** We use HumanEval (Chen et al., 2021) and MBPP (Austin et al., 2021) as the test set. **Multimodal:** We utilize the GQA (Hudson & Manning, 2019) and the image part of ScienceQA (Lu et al., 2022) datasets. Please refer to Appendix B.1 for more details.

*Table 2.* Comparison of PRINMIX and baselines on various tasks across 7B-sized models. We report the results in the format "mean(std)" with three runs for Delta-CoMe and PRINMIX.

| Method | $\alpha$ | DeepSeek-R1-Distill-Qwen | | Qwen2.5-Math-Instruct | | Qwen2.5-Coder-Instruct | | Qwen2.5-VL-Instruct | | AVG |
|---|---|---|---|---|---|---|---|---|---|---|
| | | Math500 | AIME2024 | Math500 | GSM8K | Humaneval | Mbpp | GQA | SQA | |
| Backbone | 1 | 70.6 | 16.7 | 70.6 | 84.8 | 72.0 | 80.7 | - | - | - |
| Aligned | 1 | 86.0 | 40.0 | 80.2 | 94.8 | 87.2 | 82.8 | 60.5 | 76.7 | 76.0 |
| Low-Rank | 1/16 | 72.2 | 13.3 | 59.6 | 70.3 | 84.1 | **86.2** | 0.0 | 0.0 | 48.2 |
| BitDelta | 1/16 | 1.4 | 0.0 | 71.2 | 84.0 | 83.5 | 83.9 | 0.0 | 0.3 | 40.5 |
| Delta-CoMe | 1/16 | 82.4(1.11) | 30.0(3.30) | 74.8(0.35) | 94.5(0.00) | 85.0(0.96) | 82.7(0.17) | 49.4(1.65) | 76.5(0.26) | 71.9 |
| PRINMIX | 1/16 | **82.7(0.83)** | **36.7(3.35)** | **77.7(1.03)** | **94.6(0.51)** | **85.6(0.35)** | 83.1(0.25) | **52.4(2.30)** | **79.4(0.83)** | **74.0** |

*Table 3.* Comparison of PRINMIX and baselines on various tasks across 13-14B-sized models. We report the results in the format "mean(std)" with three runs for Delta-CoMe and PRINMIX.

| Method | $\alpha$ | DeepSeek-R1-Distill-Qwen | | MetaMath | | Qwen2.5-Coder-Instruct | | LLAVA-V1.5 | | AVG |
|---|---|---|---|---|---|---|---|---|---|---|
| | | Math500 | AIME2024 | Math500 | GSM8K | Humaneval | Mbpp | GQA | SQA | |
| Backbone | 1 | 76.4 | 3.3 | 1.8 | 4.3 | 78.7 | 84.7 | - | - | - |
| Aligned | 1 | 87.4 | 40.0 | 22.6 | 71.0 | 90.2 | 85.4 | 63.3 | 72.8 | 66.6 |
| Low-Rank | 1/16 | 57.2 | 6.7 | 15.8 | 64.0 | 86.6 | **88.6** | 57.0 | 71.4 | 55.9 |
| BitDelta | 1/16 | 82.8 | 23.3 | 22.4 | 65.8 | 89.0 | 86.5 | 61.2 | **73.0** | 63.0 |
| Delta-CoMe | 1/16 | 76.5(3.38) | 24.5(6.93) | **22.9(0.12)** | 70.2(0.56) | 90.6(0.75) | 86.5(0.70) | **62.8(0.09)** | 72.3(0.20) | 63.3 |
| PRINMIX | 1/16 | **80.2(2.09)** | **31.1(3.81)** | 21.7(0.64) | **71.2(0.26)** | **91.5(0.60)** | 86.9(0.12) | 62.7(0.04) | 72.1(0.18) | **64.7** |

**Models** To ensure a comprehensive comparison, we evaluate both 7B and 13-14B models across the four tasks with various backbones, as shown in Table 1. During inference, we employ a greedy search strategy. Each model is compressed by a factor of 16 following (Ping et al., 2024) ($\alpha = 1/16$). In Appendix C.2, we report the performance of PRINMIX with different compression ratios. We further evaluate PRINMIX on three additional architectures in Section 4.3.

**Calibration Dataset** Following Delta-CoMe (Ping et al., 2024), PRINMIX randomly samples 128 examples, each containing 2048 tokens, from the C4 training set as the calibration dataset. In Appendix C.3, we analyze the sensitivity of the calibration dataset by varying its domains and the number of examples. The results show that PRINMIX performs consistently well across different calibration dataset configurations.

**Baselines** We compare PRINMIX with three baselines: SVD-based low-rank compression (Ryu et al., 2023), BitDelta (Liu et al., 2024), and Delta-CoMe (Ping et al., 2024). All methods are evaluated using NVIDIA L20 GPUs.

### 4.2. Main Results

Tables 2 and 3 present the results of PRINMIX on both the 7B and 13-14B models across four tasks, in comparison to the baselines. Notably, PRINMIX demonstrates superior overall performance on both the 7B and 13-14B models, surpassing the best baseline, Delta-CoMe, by an

When analyzing the various tasks, we observe that PRIN-

MIX exhibits more pronounced improvements in challenging scenarios characterized by a significant performance gap between the baseline methods and the aligned model. This is particularly evident in reasoning-intensive benchmarks, such as AIME2024, as well as in multimodal tasks utilizing 7B backbones. For instance, PRINMIX surpasses the previous state-of-the-art model, Delta-CoMe, by 22.3% on the 7B model and by 26.9% on the 14B model. Further analysis reveals that these models display larger norms for $\Delta\mathbf{W}$. Specifically, the median norm of DeepSeek-R1-Distill-Qwen-7B and Qwen2.5-VL-Instruct is 6.5 and 10.3 times that of Qwen2.5-Coder-Instruct, with corresponding values of 26.13 and 41.45 compared to 4.02, respectively. In this context, baseline methods struggle to achieve optimal solutions due to their empirical nature. In contrast, PRIN-MIX directly optimizes quantization error from a mathematical perspective, enabling it to fully leverage its strengths in demanding tasks. However, on tasks where baselines already achieve near-lossless accuracy, such as MBPP and HumanEval on the 7B backbone, PRINMIX performs comparably to the best baseline. In these scenarios, the norm of $\Delta\mathbf{W}$ is relatively small and can be easily compressed, leading to a ceiling effect: $\Delta\mathbf{W}$ can be quantized almost losslessly by existing baselines, leaving little room for further improvement.

### 4.3. Generalization to Diverse Models

To demonstrate the generalization of PRINMIX, we conduct additional experiments on the recent Qwen3 model family (Qwen3-4B / Qwen3-4B-Thinking-2507) (Team, 2025), and two new architectures: (1) a Mixture-of-Experts model (OLMoE-1B-7B-0924 / OLMoE-1B-7B-

*Table 4.* Performance of PRINMIX on MoE, encoder-decoder, and recent model architectures. OLMoE-1B-7B-0924-Instruct is aligned from OLMoE-1B-7B-0924 (MoE); Flan-T5-XL is aligned from T5-v1.1-XL (encoder-decoder); Qwen3-4B-Thinking-2507 is aligned from Qwen3-4B (recent decoder-only). AVG is computed over all six benchmarks.

| Method | $\alpha$ | OLMoE-1B-7B-0924-Instruct | | Flan-T5-XL | | Qwen3-4B-Thinking-2507 | | AVG |
|---|---|---|---|---|---|---|---|---|
| | | GSM8K | MMLU | MMLU | IFEval | Math500 | AIME2024 | |
| Aligned | 1 | 46.3 | 51.7 | 48.7 | 17.0 | 80.2 | 23.3 | 44.5 |
| Low-Rank | 1/16 | 26.5 | 49.2 | 47.8 | 15.9 | 78.4 | 16.7 | 39.1 |
| BitDelta | 1/16 | 38.4 | **52.0** | 48.7 | 17.0 | 80.2 | 16.7 | 42.2 |
| Delta-CoMe | 1/16 | 43.1 | 51.4 | 48.3 | 16.1 | 78.4 | 20.0 | 42.9 |
| PRINMIX | 1/16 | **44.1** | 51.8 | **48.8** | **17.7** | **81.0** | **23.3** | **44.5** |

*Table 5.* Performance across different $f_{max}$. We report the results in the format "mean(std)" with three runs.

| Method | $f_{max}$ | DeepSeek-R1-Distill-Qwen-14B | | AVG |
|---|---|---|---|---|
| | | Math500 | AIME2024 | |
| Delta-CoMe | - | 76.5(3.38) | 24.5(6.93) | 50.5 |
| | 2 | **80.7(1.75)** | **33.3(3.35)** | **57.0** |
| | 3 | 79.9(1.53) | 30.0(8.83) | 55.0 |
| PRINMIX | 4 | 80.2(2.09) | 31.1(3.81) | 55.7 |
| | 5 | 79.5(0.99) | **33.3(6.65)** | 56.4 |
| | 6 | 79.5(2.21) | **33.3(3.35)** | 56.4 |

*Table 6.* Ablation of RTC. We report the results in the format "mean(std)" with three runs.

| | LLAVA-V1.5 | | DeepSeek-R1-Distill-Qwen-14B | | AVG |
|---|---|---|---|---|---|
| | GQA | SQA | Math500 | AIME2024 | |
| Delta-CoMe | **62.8(0.09)** | **72.3(0.20)** | 76.5(3.38) | 24.5(6.93) | 59.0 |
| PRINMIX | 62.7(0.04) | 72.1(0.18) | **80.2(2.09)** | **31.1(3.81)** | 61.5 |
| PRINMIX (W/O RTC) | **62.8(0.02)** | 72.2(0.05) | 78.2(0.28) | 27.5(3.81) | 60.2 |

*Table 7.* Performance comparison between Delta Compression and LoRA. Aligned is full fine-tuned model. For PRINMIX, we report the results in the format "mean(std)" with three runs.

| Method | $\alpha$ | Code | | Math | | AVG |
|---|---|---|---|---|---|---|
| | | Humaneval | Mbpp | Math500 | GSM8K | |
| Backbone | 1 | 24.4 | 46.0 | 3.8 | 14.7 | 22.2 |
| Aligned | 1 | 46.3 | 48.9 | 14.6 | 58.3 | 42.0 |
| LoRA | 1/16 | 34.1 | 47.7 | 9.4 | 50.9 | 35.5 |
| PRINMIX | 1/16 | **43.3(0.60)** | **50.2(0.82)** | **13.5(0.76)** | **56.1(0.82)** | 40.8 |
| PRINMIX-LoRA | 1/64 | 34.1(1.64) | 47.7(0.85) | 9.1(0.50) | 49.7(0.30) | 35.2 |
| PRINMIX | 1/64 | **39.4(1.56)** | **50.0(1.10)** | **11.1(0.95)** | **52.9(0.57)** | **38.4** |

0924-Instruct) (Muennighoff et al., 2024), (2) an encoder-decoder model (T5-v1.1-XL (Raffel et al., 2023) / Flan-T5-XL (Chung et al., 2022)). As shown in Table 4, PRINMIX generalizes well across all three settings and outperforms all baselines. It achieves an average score of 44.5 across the six reported benchmarks, matching the aligned model on average (44.5).

### 4.4. Ablation of $f_{max}$

In PRINMIX, we set a hyperparameter termed $f_{max}$ to constrain the number of active bit-widths during quantization. This section examines the performance of PRINMIX under varying values of $f_{max}$. As shown in Table 5, PRINMIX consistently achieves better performance than Delta-CoMe across all settings, indicating that PRINMIX is insensitive to the choice of $f_{max}$. In the main experiment, we set $f_{max}$ to 4 to be consistent with Delta-CoMe.

### 4.5. Ablation of RTC

We assess the necessity of RTC in Table 6. Overall, RTC consistently enhances our method, yielding an average per-

formance improvement of 2.2%. The results indicate that mitigating the deviation in the quantization loss of $\mathbf{U}$ enables PRINMIX to retain more information from $\Delta \mathbf{W}$. The importance of RTC is particularly pronounced in challenging tasks; for instance, it improves performance by 13.1% on the AIME2024 task. This improvement can be attributed to the more substantial quantization errors associated with quantizing $\mathbf{V}$ in these cases, thereby highlighting the importance of reconstruction target correction in challenging settings.

## 5. Analyses

### 5.1. Delta Compression vs. Delta-Tuning

Delta compression decomposes the delta weights of a fully fine-tuned model into low-rank and low-bit representations, thereby reducing storage and inference costs. Delta-tuning methods, such as LoRA, are closely related to delta compression but primarily aim to reduce the training costs of LLMs while achieving performance comparable to that of full fine-tuning. However, in various tasks—particularly more complex ones like code and math tasks—delta-tuning methods tend to underperform full fine-tuning (Biderman et al., 2024). This suggests that relying solely on delta-tuning may be insufficient.

In this section, we train the DeepSeek-LLM-7B-Base (DeepSeek-AI, 2024) on math and code tasks using both LoRA and full fine-tuning. We subsequently apply

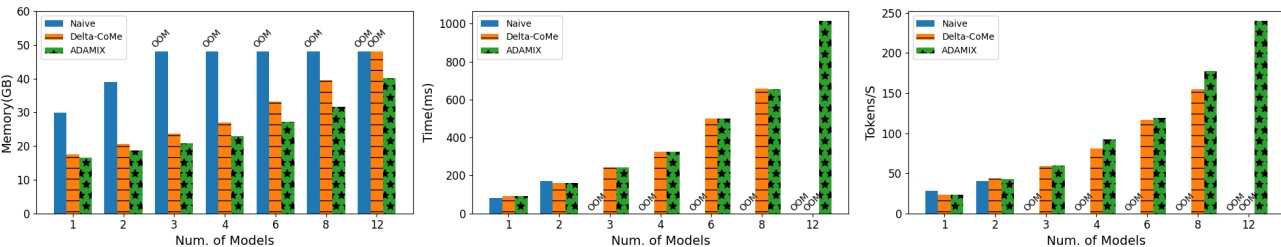

*Figure 3.* End-to-end decoding latency evaluation with varying numbers of deployed models using Qwen2.5-7B variants. (Left) Decoding memory usage. (Middle) Prefill time. (Right) Generation speed.

PRINMIX to the delta weights of the fully fine-tuned model and LoRA. Additional experimental details can be found in Appendix B.2. Table 7 presents a comparison of PRIN-MIX with LoRA. PRINMIX consistently outperforms LoRA across all tasks. When $\alpha = 1/16$, PRINMIX achieves an average score of 40.8, which is close to the aligned model's score of 42.0, representing a 14.9% improvement over LoRA.

In addition to full fine-tuning model, PRINMIX can effectively compress the LoRA model as well. As illustrated in Table 7, PRINMIX-LoRA demonstrates a performance degradation of only 0.3 compared to the LoRA, while further compressing LoRA with a ratio of 4 ($\alpha = 1/64$). Notably, Baselines like BitDelta and Delta-CoMe cannot apply to LoRA. BitDelta directly quantizes $\Delta\mathbf{W}$ to 1 bit without employing any low-rank approximation. Consequently, it cannot effectively utilize the low-rank properties inherent in LoRA. For Delta-CoMe, the empirically determined mixed-precision scheme is fixed and does not offer a clear method for allocating mixed-precision at other compression ratios. In contrast, PRINMIX allows compression of $\Delta\mathbf{W}$ to arbitrary ratios, making it more flexible and practically advantageous. Although PRINMIX-LoRA performs well on top of LoRA, it underperforms PRINMIX at the same compression ratio ($\alpha = 1/64$). This further highlights the significance of delta compression in comparison to delta-tuning.

### 5.2. Inference Speed and Memory Cost

Following the setup of Liu et al. (2024), we evaluate the end-to-end decoding latency of Qwen2.5-7B variants using a single L20 GPU. As shown in Figure 3, we consider the setting where each deployed model receives one distinct request simultaneously—e.g., 12 deployed models correspond to a batch size of 12- with latency evaluation in three perspectives: (1) Memory Usage: This one measures peak GPU memory usage during concurrent inference, accounting for both model parameters and activation storage. (2) Prefill Time: This part focuses on the time the models take to process user-input prompts. Each request contains 512 input tokens, and we report the time (in ms) the model takes to handle them. (3) Generation Speed: This part evaluates

*Table 8.* Comparison between direct aligned-model quantization (DirectC) and PRINMIX on Qwen2.5-Math-7B-Instruct using GPTQ. DirectC quantizes the entire aligned model, whereas PRIN-MIX uses a shared quantized base model and compressed delta weights for multi-tenant deployment. "—" denotes not applicable.

| Method | Storage | | | Performance | | |
|---|---|---|---|---|---|---|
| | Aligned Bit | Base Bit | Delta Bit | Math500 | GSM8K | Avg Score |
| DirectC | 4 | — | — | 78.2 | 94.8 | 86.5 |
| DirectC | 2 | — | — | 2.4 | 3.4 | 2.9 |
| PRINMIX | — | 4 | 1 | 77.4 | 92.4 | 84.9 |
| PRINMIX | — | 8 | 1 | 76.0 | 94.4 | 85.2 |

how quickly the model generates output tokens (tokens/s) for each request. Since the prefill time already measures prompt processing, each request starts from the "[BOS]" token and generates 512 tokens sequentially.

As shown in Figure 3 (left), a single GPU can deploy only two aligned models simultaneously. In contrast, it can support up to 8 and 12 models concurrently for Delta-CoMe and PRINMIX, respectively. This enhancement is attributable to the fact that, as the number of models increases, both methods necessitate only the additional deployment of compressed delta weights, thereby significantly reducing memory overhead. Notably, while Delta-CoMe exhausts GPU memory at 12 models, PRINMIX does not. Our further analysis indicates that PRINMIX typically employs fewer ranks, namely allocates a greater number of singular vectors with a bid-width of 0, thereby enhancing the GPU memory utilization efficiency.

For the end-to-end decoding latency illustrated in Figure 3 (middle, right), we find that Delta-CoMe and PRINMIX introduce overhead to Naive when the number of deployed model is small. However, Delta-CoMe and PRINMIX scale better and effectively translate the saved GPU memory into improved decoding latency. In contrast, the Naive approach quickly encounters out-of-memory issues. Furthermore, PRINMIX exhibits a superior generation speed compared to Delta-CoMe at scale, while the prefill times for both methods remain comparable. In Appendix C.7, we conduct more latency evaluation under varying arrival rates and request distributions following DeltaZip (Yao et al., 2024).

## 5.3. Comparison with Direct Model Quantization

A natural question is how PRINMIX compares with directly quantizing the aligned model (DirectC). Table 8 reports the storage cost and performance on Qwen2.5-Math-7B-Instruct, using GPTQ as the quantization backend. Although PRINMIX achieves slightly lower accuracy than DirectC under 4-bit quantization, it offers distinct advantages in multi-tenant serving scenarios. First, given $N$ models to deploy, PRINMIX requires only one shared quantized base model together with $N$ sets of 1-bit delta parameters, whereas DirectC requires deploying $N$ separate 4-bit quantized aligned models. As $N$ increases, this storage advantage becomes increasingly significant. Second, switching between models in PRINMIX only involves loading or offloading the 1-bit delta parameters, which incurs substantially lower I/O overhead than loading or offloading full 4-bit quantized aligned models as in DirectC. Therefore, although PRINMIX is slightly less accurate than DirectC for a single model, it is a more suitable compression and quantization solution for the target multi-tenant setting when both storage efficiency and serving overhead are taken into account.

## 5.4. Analyzing Quantization Error

To better understand the difference between various delta compression methods, we compute the quantization error on Qwen2.5-Math-7B-Instruct model as defined in Equation (1). Since outliers play a critical role in model compression (Dettmers et al., 2023; Lin et al., 2024), we also report the average error for the top 1% of activations with the largest absolute values in the aligned model, categorizing them as outliers. As different layers contribute differently to the final output (Wu et al., 2024), we categorize the first 9 layers, layers 9 to 17, and the last 10 layers as low, mid, and high groups, respectively, and report the average error of each group. See Table 21 of Appendix C.9 for more details.

As demonstrated in Table 9, PRINMIX consistently exhibits lower overall quantization error compared to all baseline methods, attributable to its inherent objective of minimizing the quantization error. In the mid layers, PRINMIX shows a slightly higher error than the BitDelta, with values of 0.66 versus 0.61 for all activations and 1.12 versus 1.08 for outlier activations, respectively. However, it is important to note that since BitDelta is an empirical method, it cannot guarantee low quantization error across all layers. For example, in the high layers, BitDelta exhibits significantly higher error rates compared to PRINMIX, with values of 21.51 versus 6.81 for all activations and 3162.58 versus 426.20 for outlier activations, respectively. These experiments further illustrate that PRINMIX effectively reduces quantization error, thereby preserving the information contained in the delta weights as much as possible. In Appendix C.8, we visualize the bit allocation results of PRINMIX across

*Table 9*. Average quantization error ($\times$ 1e2) on Qwen2.5-Math-7B-Instruct model with Eq. (1)."Low", "Mid", and "High" denote the first 9 layers, layers 9 to 17, and the last 10 layers, respectively. "All" and "Out" denote the average error across all activations and the average error of the top 1% of activations.

| | Low | | Mid | | High | |
|---|---|---|---|---|---|---|
| | All | Out | All | Out | All | Out |
| Low-Rank | 1.82 | 3.67 | 1.50 | 2.84 | 21.12 | 1890.34 |
| BitDelta | 2.18 | 2.81 | **0.61** | **1.08** | 21.51 | 3162.58 |
| Delta-CoMe | 0.76 | 1.79 | 0.75 | 1.33 | 7.54 | 470.82 |
| PRINMIX | **0.66** | **1.46** | 0.66 | 1.12 | **6.81** | **426.20** |

different weight types and layers using the Qwen2.5-Math-7B-Instruct model.

## 6. Conclusion

In this study, we present PRINMIX, an adaptive mixed-precision delta compression framework aimed at minimizing quantization error in the SVD space without introducing additional assumptions. PRINMIX offers a formal analysis of when mixed-precision is advantageous for delta compression under the GPTQ-style objective and provides a practical quantization solution that involves solving a 0/1 linear integer programming problem and employing a reconstruction target correction method. PRINMIX outperforms all baseline delta compression methods across four distinct downstream tasks, including reasoning, math, code, and multimodal, utilizing eight widely adopted aligned LLMs with backbone pre-trained models, including Qwen2.5, Qwen2.5-Math, Qwen2.5-Coder, and LLaMA2. Moreover, PRINMIX significantly reduces deployment costs by minimizing memory overhead and accelerating inference. We believe that PRINMIX provides considerable theoretical and practical value, particularly in scenarios involving multi-tenant deployments.

## Impact Statement

PRINMIX significantly reduces hardware requirements and computational costs for serving multiple finetuned models, thereby enabling smaller entities to deploy advanced large language models more feasibly. Additionally, it lowers power consumption and reduces the carbon emissions associated with LLM deployment. Despite PRINMIX's demonstrated improvements over baseline methods in reducing the performance gap between compressed and aligned models, it is important to note that PRINMIX remains a lossy compression method for certain tasks. We believe this is an important consequence and encourage future research to further minimize this performance gap, particularly in tasks where performance degradation is substantial.

## Acknowledgements

This project was supported by National Natural Science Foundation of China (No. 62306132), Guangdong Basic and Applied Basic Research Foundation (No. 2025A1515011564), Natural Science Foundation of Shanghai (No. 25ZR1402136) and Shanghai Science and Technology Committee under Grant No. 24511103900. We thank the anonymous reviewers for their insightful feedback on this work.

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

# A. Formula Derivation

## A.1. V Hessian Matrix

$$
\begin{aligned}
&d_{\hat{\mathbf{V}}}^2 \left\| \mathbf{U}\mathbf{\Sigma}\mathbf{V}X - \mathbf{U}\mathbf{\Sigma}\hat{\mathbf{V}}X \right\|_F^2 \\
&= 2tr(\mathbf{U}\mathbf{\Sigma}d\hat{\mathbf{V}}XX^T d\hat{\mathbf{V}}^{\mathbf{T}}\mathbf{\Sigma}^{\mathbf{T}}\mathbf{U}^{\mathbf{T}}) \\
&= 2tr(\mathbf{\Sigma}^{\mathbf{T}}\mathbf{U}^{\mathbf{T}}\mathbf{U}\mathbf{\Sigma}d\hat{\mathbf{V}}XX^T d\hat{\mathbf{V}}^{\mathbf{T}}) \\
&= 2(d\,\mathrm{vec}(\hat{V}^T))^T(\mathbf{\Sigma}^{\mathbf{T}}\mathbf{\Sigma} \otimes XX^T)(d\,\mathrm{vec}(\hat{V}^T)) \\
&= 2(d\,\mathrm{vec}\hat{V})^T(\mathbf{\Sigma}^{\mathbf{T}}\mathbf{\Sigma} \otimes XX^T)(d\,\mathrm{vec}(\hat{V})) \\
&\Rightarrow \mathbf{H}^{\mathbf{V}} = 2\mathbf{\Sigma}^{\mathbf{T}}\mathbf{\Sigma} \otimes XX^T \\
&\Rightarrow \mathbf{H}_i^{\mathbf{V}} = 2\Sigma_{ii}^2 \cdot XX^T
\end{aligned}
\tag{9}
$$

Here $\otimes$ denotes the Kronecker product.

## A.2. U Hessian Matrix

$$
\begin{aligned}
&d_{\hat{\mathbf{U}}}^2 \left\| \mathbf{U}\mathbf{\Sigma}\hat{\mathbf{V}}X - \hat{\mathbf{U}}\mathbf{\Sigma}\hat{\mathbf{V}}X \right\|_F^2 \\
&= d\hat{\mathbf{U}}\mathbf{\Sigma}\hat{\mathbf{V}}XX^T\hat{\mathbf{V}}^{\mathbf{T}}\mathbf{\Sigma}^{\mathbf{T}}d\hat{\mathbf{U}}^{\mathbf{T}} \\
&= X^T\hat{\mathbf{V}}^{\mathbf{T}}\mathbf{\Sigma}^{\mathbf{T}}d\hat{\mathbf{U}}^{\mathbf{T}}d\hat{\mathbf{U}}\mathbf{\Sigma}\hat{\mathbf{V}}X \\
&= (d\,\mathrm{vec}\hat{U})^T\mathbf{K}_{\mathbf{rh_{out}}}(\mathbf{I} \otimes \mathbf{\Sigma}\hat{\mathbf{V}}XX^T\hat{\mathbf{V}}^{\mathbf{T}}\mathbf{\Sigma}^{\mathbf{T}})\mathbf{K}_{\mathbf{h_{out}r}}(d\,\mathrm{vec}\hat{U}) \\
&= 2(d\,\mathrm{vec}\hat{U})^T(\mathbf{I} \otimes \mathbf{\Sigma}\hat{\mathbf{V}}XX^T\hat{\mathbf{V}}^{\mathbf{T}}\mathbf{\Sigma}^{\mathbf{T}})(d\,\mathrm{vec}\hat{U}) \\
&\Rightarrow \mathbf{H}_i^{\mathbf{U}} = \mathbf{H}^{\mathbf{U}} = 2\mathbf{\Sigma}\hat{\mathbf{V}}XX^T\hat{\mathbf{V}}^{\mathbf{T}}\mathbf{\Sigma}^{\mathbf{T}}
\end{aligned}
\tag{10}
$$

Here $\mathbf{K}_{\mathbf{h_{out}r}}$ is the commutation matrix, and $\mathbf{K}_{\mathbf{h_{out}r}}^{-1} = \mathbf{K}_{\mathbf{rh_{out}}}$.

## A.3. Detailed Derivation Process for new U

$$
\begin{aligned}
&d_{\tilde{\mathbf{U}}} \left\| \mathbf{U}\mathbf{\Sigma}\mathbf{V}X - \tilde{\mathbf{U}}\mathbf{\Sigma}\hat{\mathbf{V}}X \right\|_F^2 \\
&= 2tr(d\tilde{\mathbf{U}}\mathbf{\Sigma}\hat{\mathbf{V}}X(\tilde{\mathbf{U}}\mathbf{\Sigma}\hat{\mathbf{V}}X - \mathbf{U}\mathbf{\Sigma}\mathbf{V}X)^T) \\
&= 2tr(\mathbf{\Sigma}\hat{\mathbf{V}}X(\tilde{\mathbf{U}}\mathbf{\Sigma}\hat{\mathbf{V}}X - \mathbf{U}\mathbf{\Sigma}\mathbf{V}X)^T d\tilde{\mathbf{U}}) \\
&\Rightarrow \frac{\partial \mathbb{L}}{\partial \tilde{\mathbf{U}}} = (\tilde{\mathbf{U}}\mathbf{\Sigma}\hat{\mathbf{V}}X - \mathbf{U}\mathbf{\Sigma}\mathbf{V}X)X^T\hat{\mathbf{V}}^{\mathbf{T}}\mathbf{\Sigma}^{\mathbf{T}}
\end{aligned}
\tag{11}
$$

By setting the gradient of the loss to zero, PRINMIX gets the corrected $\tilde{\mathbf{U}}$ as follow:

$$
\begin{aligned}
&\frac{\partial \mathbb{L}}{\partial \tilde{\mathbf{U}}} = (\tilde{\mathbf{U}}\mathbf{\Sigma}\hat{\mathbf{V}}X - \mathbf{U}\mathbf{\Sigma}\mathbf{V}X)X^T\hat{\mathbf{V}}^{\mathbf{T}}\mathbf{\Sigma}^{\mathbf{T}} = 0 \\
&\Rightarrow \tilde{\mathbf{U}} = \mathbf{U}\mathbf{\Sigma}\mathbf{V}XX^T\hat{\mathbf{V}}^{\mathbf{T}}\mathbf{\Sigma}^{\mathbf{T}}(\mathbf{\Sigma}\hat{\mathbf{V}}XX^T\hat{\mathbf{V}}^{\mathbf{T}}\mathbf{\Sigma}^{\mathbf{T}})^{-1}
\end{aligned}
\tag{12}
$$

# B. Experiments Setup

## B.1. Main Experiments

We evaluate our methods across models in Table 1 on four distinct tasks: math, reasoning, code generation, and multimodal. These tasks encompass a vast array of current directions based on fine-tuning with open-source LLMs.

• **Math.** We use the GSM8K (Cobbe et al., 2021) and Math500 (Lightman et al., 2023) datasets as the test set. We follow the prompt format of WizardMath (Luo et al., 2025) and set the maximum generation length to 1024. The evaluation metric is accuracy, determined by comparing the model-generated solution to the ground truth.

● **Reasoning.** We use the Math500 and AIME2024 datasets as the test set. For the reasoning prompt of AIME2024, we follow with (Naman Jain & et al., 2024). The maximum length of both tasks is set to 8192. The evaluation metric is accuracy, determined by comparing the model-generated solution to the ground truth.

● **Code Generation.** We use two widely used datasets as the test set: HumanEval (Chen et al., 2021) and MBPP (Austin et al., 2021). We follow the Magicoder (Wei et al., 2024) evaluation framework for HumanEval and adopt EvalPlus (Liu et al., 2023a) for MBPP. The evaluation metric is the pass rate (pass@1), which measures whether the code generated in a single attempt successfully passes the test cases.

● **Multimodal.** We utilize the GQA (Hudson & Manning, 2019) and the image part of ScienceQA (Lu et al., 2022) datasets, both commonly used for evaluating VLM performance, as our test set. We adopt lmms-eval (Zhang et al., 2024) to evaluate both tasks. The evaluation metric is accuracy, which measures whether the model selects the correct option.

### B.2. Delta Compression vs. Delta-Tuning

Specifically, we set the LoRA rank to 128 and the scale factor to 256, training LoRA for all model parameters for 3 epochs using a cosine schedule with a peak learning rate of 4e-5 and a warm-up ratio of 0.1, using model deepseek-llm-7b-base (DeepSeek-AI, 2024). We randomly sample 50k training examples from MetaMathQA (Yu et al., 2023) and Magicoder-Evol-Instruct (Wei et al., 2024) for the math and code tasks, respectively. To ensure a fair comparison, we fine-tune all model parameters using the same datasets as those used for LoRA training. We then apply PRINMIX to both math and code finetuned LLMs.

## C. More Experiments

### C.1. Analyzing the Different Quantization Schemes in U

In this section, we investigate the effect of applying different quantization schemes to $\mathbf{U}$ in order to assess the necessity of mixed-precision. Our evaluation is conducted on Qwen2.5-Math-7B-Instruct. The results show that the choice of quantization strategy for $\mathbf{U}$ has a relatively limited effect on overall performance compared with the overall mixed-precision design, and PRINMIX setting remains the strongest among the tested variants. As shown in Table 10, "x-bit" denotes quantization of $\mathbf{U}$ with x-bit precision. The "PRINMIX-row" setting applies the optimization model to determine the scheme and performs quantization in a row-wise manner,

*Table 10.* We evaluate the performance of various quantization schemes applied to $\mathbf{U}$ on Qwen2.5-Math-7B-Instruct. Here, "x-bit" denotes quantization of $\mathbf{U}$ at x-bit precision. The "PRINMIX-row" setting refers to applying the optimization model to determine the scheme and performing quantization in a row-wise manner, whereas "PRINMIX " indicates employing the same quantization scheme used for $\mathbf{V}$, with quantization carried out column by column. For internal consistency, the PRINMIX row at $\alpha = 1/16$ reuses the corresponding main-results values in Table 2.

|  | $\alpha$ | Math500 | GSM8K | AVG |
|---|---|---|---|---|
| **U**(2bit),**V**(PRINMIX) | 1/16 | 76.8 | 93.6 | 85.2 |
| **U**(3bit),**V**(PRINMIX) | 1/16 | 75.6 | 93.4 | 84.5 |
| **U**(PRINMIX-row),**V**(PRINMIX) | 1/16 | 75.2 | 93.6 | 84.4 |
| PRINMIX | 1/16 | 77.7 | 94.6 | 86.2 |

whereas "PRINMIX" adopts the same quantization scheme as $\mathbf{V}$ and conducts quantization column by column. The performance differences across schemes remain relatively limited: compared with PRINMIX, the fixed 2-bit quantization is lower by 1.0 average point, while "PRINMIX-row" is lower by 1.8 average points. These results suggest that the choice of quantization strategy for $\mathbf{U}$ has a secondary effect on overall performance.

To provide a supplemental validation under a more complete objective, we further evaluate an approximate joint ILP formulation that incorporates a column error term for $\mathbf{U}$ under the assumption that a paired $\mathbf{U}$-column and $\mathbf{V}$-row share the same bit-width.

As shown in Table 11, this approximate joint objective underperforms PRINMIX on Math500 by 2.5 points and matches it on AIME2024 while increasing optimization complexity. Without the shared-bit coupling assumption, forming an exact joint objective is computationally prohibitive because the reconstruction

*Table 11.* Supplemental validation of an approximate joint ILP formulation on DeepSeek-R1-Distill-Qwen-7B. This table reports Math500 and AIME2024 only and is not directly comparable to Table 10, which uses Qwen2.5-Math-7B-Instruct.

| Setting | $\alpha$ | Math500 | AIME2024 | AVG |
|---|---|---|---|---|
| Joint ILP | 1/16 | 80.2 | 36.7 | 58.5 |
| PRINMIX | 1/16 | 82.7 | 36.7 | 59.7 |

error for $\mathbf{U}$ depends on the quantized assignment of $\mathbf{V}$, whose feasible assignments grow combinatorially. These results provide supplemental evidence for the design choice of optimizing $\mathbf{V}$ only.

## C.2. Ablation of Compression Ratio

To show that PRINMIX can apply to arbitrary compression ratios, we evaluate Qwen2.5-Math-7B-Instruct at four compression ratios, as shown in Table 12. The performance of PRINMIX generally degrades as compression becomes more aggressive.

This is expected, as a higher compression ratio indicates a reduced capacity of the quantized model to preserve information from the original model. Notably, baselines like Bit-Delta and Delta-CoMe cannot apply to other compression ratios except $\alpha =$1/16. BitDelta quantizes $\Delta\mathbf{W}$ to a fixed 1 bit, resulting in a constant compression ratio. For Delta-CoMe, the empirically determined mixed-precision scheme is fixed and does not offer a clear method for allocating mixed-precision at other compression ratios. In contrast,

*Table 12.* Performance of PRINMIX under different compression ratios $1/\alpha$. For internal consistency, the overlapping $\alpha = 1/16$ PRINMIX entry reuses the corresponding main-results values in Table 2.

| $\alpha$ | DeepSeek-R1-Distill-Qwen-7B | | Qwen2.5-Math-7B-Instruct | | |
|---|---|---|---|---|---|
| | Math500 | AIME2024 | Math500 | GSM8K | AVG |
| 3/16 | 86.4 | 36.7 | 77.2 | 95.6 | 74.0 |
| 2/16 | 85.8 | 33.3 | 77.4 | 95.1 | 72.9 |
| 1/16 | 82.7 | 36.7 | 77.7 | 94.6 | 72.9 |
| 1/32 | 76.8 | 26.7 | 73.4 | 91.6 | 67.1 |

PRINMIX enables the compression of $\Delta\mathbf{W}$ to arbitrary ratios, offering greater flexibility and broader applicability.

*Table 13.* The performance of PRINMIX to quantize Qwen2.5-Math-7B-Instruct with different number of calibration data.

| Calibration Size | Math500 | GSM8K | AVG |
|---|---|---|---|
| 16 | 76.4 | 94.5 | 85.5 |
| 32 | 76.2 | **95.1** | 85.7 |
| 64 | 76.8 | 94.3 | 85.6 |
| 128 | **77.7** | 94.6 | **86.2** |
| 256 | 76.0 | 94.1 | 85.1 |

*Table 14.* The performance of PRINMIX to quantize Qwen2.5-Math-7B-Instruct using calibration data drawn from C4 and Wikitext2.

| | Math500 | GSM8K | AVG |
|---|---|---|---|
| C4 | **77.7** | 94.6 | **86.2** |
| Wikitext2 | 76.6 | **94.8** | 85.7 |
| MetaMath | 75.4 | 93.6 | 84.5 |

## C.3. Ablation of Calibration Dataset

Since PRINMIX is a calibration-dependent method, to verify its robustness to calibration, we examine different sizes and domains of the calibration dataset for quantizing Qwen2.5-Math-7B-Instruct. For calibration on domains, each calibration set contains 128 randomly sampled sequences of length 2048. Due to the insufficient number of sequences of this length in the MetaMathQA dataset, we concatenate multiple question–answer pairs in a few-shot format. To examine the effect of dataset size on calibration, we vary the number of calibration samples on the C4 dataset from 16 to 256. The results in Tables 13 and 14 demonstrate that PRINMIX performs well on all calibration setups. The average performance gap is within 2.0%, confirming PRINMIX 's robustness.

## C.4. Numerical Stability of RTC

The RTC step (Eq. 7) requires inverting $\mathbf{\Sigma}\hat{\mathbf{V}}XX^T\hat{\mathbf{V}}^\mathbf{T}\mathbf{\Sigma}^\mathbf{T}$. In practice, we compute this inverse via Cholesky decomposition. When the Cholesky decomposition fails (e.g., the matrix is not positive definite), we add a diagonal regularization term $\lambda\mathbf{I}$ to $\mathbf{\Sigma}\hat{\mathbf{V}}XX^T\hat{\mathbf{V}}^\mathbf{T}\mathbf{\Sigma}^\mathbf{T}$ before decomposition, where $\lambda = \max\left(|\min \text{ eigenvalue}|, 10^{-4}\right)$. This offsets negative eigenvalues and provides a small numerical floor when the spectrum is already close to positive semidefinite.

*Table 15.* Per-layer condition number statistics (percentiles) of $\mathbf{\Sigma}\hat{\mathbf{V}}XX^T\hat{\mathbf{V}}^\mathbf{T}\mathbf{\Sigma}^\mathbf{T}$ across all transformer blocks.

| Model | 5% | 25% | 50% | 75% | 95% |
|---|---|---|---|---|---|
| DeepSeek-R1-Distill-Qwen-7B | 388.8 | 1874.6 | 6404.1 | 47827.0 | 1769593.3 |
| Qwen3-4B-Thinking-2507 | 133.8 | 673.7 | 3868.9 | 13891.3 | 156820.7 |

Table 15 reports the distribution of per-layer condition numbers across all transformer blocks for two representative models.

Despite the moderate-to-large condition numbers at higher percentiles, the diagonal regularization yields numerically stable inversion in our experiments. Furthermore, the calibration size ablation in Table 13 demonstrates that PRINMIX's performance is robust to the choice of calibration set size, suggesting that the matrix estimates are sufficiently stable for this procedure across the tested configurations.

### C.5. Time For Quantization

*Table 16.* Time cost (in seconds) for "Simulation", "Optimization", and "Quantization" for one transformer block on the Qwen2.5-Math-7B-Instruct model, which consists of 28 blocks.

|  |  | Simulation | Optimization | Quantization | Total |
|---|---|---|---|---|---|
| PRINMIX | Q_proj | 3.1 |  | 1.0 |  |
|  | K_proj | 3.1 | 17.5 | 1.0 |  |
|  | V_proj | 3.1 |  | 1.0 |  |
|  | O_proj | 4.0 | 11.5 | 1.5 | 134.3 |
|  | Up_proj | 3.3 | 20.5 | 2.8 |  |
|  | Gate_proj | 3.3 |  | 2.8 |  |
|  | Down_proj | 19.7 | 24.1 | 11.0 |  |

In this section, we evaluate the quantization time of PRINMIX within a single transformer block. PRINMIX determines the mixed-precision quantization strategy by minimizing quantization loss, formulated as a 0/1 integer linear programming problem. To clarify the computational overhead, we decompose the quantization time into three components. The first is "simulation time", which reflects the cost of estimating quantization loss under different bit-widths. The second is "optimization time", incurred when solving the 0/1 integer linear programming problem. The third is the "quantization time" itself, representing the cost of quantizing each linear layer according to the selected strategy. The corresponding results for one transformer block of Qwen2.5-Math-7B-Instruct, which contains 28 blocks in total, are summarized in Table 16. As shown in Table 16, most of the quantization time is spent on "simulation" and "optimization". For example, the "simulation" and "optimization" times in Q, K, and V_proj are 9.3s and 17.5s, respectively, while the quantization time only takes 3.0s. Specifically, simulation time increases with the number of columns, while optimization time grows with the number of rows.

Overall, although PRINMIX requires 1.1 hours for 7B models and 2.4 hours for 14B models on a single L20 GPU, this is acceptable.

**Accelerating the Optimization Time.** As illustrated in Table 16, solving the optimization problem constitutes the primary bottleneck, dominating the total quantization time. To mitigate this overhead, we propose accelerating the ILP solving process through two main avenues: ① increasing the ILP solving speed, and ② constraining the solution space. Specifically, regarding ①, while we employ open-source solvers in this study, switching to a commercial solver like COPT (Ge et al., 2023) can yield a 6× speedup. Furthermore, for ②, the process can be accelerated by limiting the number of candidate bit-widths (e.g., reducing candidates from 8 to 4). Our experiments in Appendix C.6 demonstrate that PRINMIX performs consistently well with different numbers of candidate bit-widths.

These methods significantly enhance PRINMIX's efficiency in practical deployment. Leveraging the acceleration strategies yields a 4x speedup in solving the optimization problem. Consequently, the quantization process for PRINMIX only takes 0.5 hours for 7B models and 1.2 hours for 14B models, representing a 50% reduction in temporal overhead.

**Scalability to Larger Models.** Table 17 reports the per-block quantization time of PRINMIX on 7B, 14B, and 70B models. The total quantization time for 70B models is approximately 8.4 hours on a single L40 GPU, which constitutes a one-time offline cost that is substantially lower than the fine-tuning overhead of 70B-scale models. Simulation time scales with hidden and intermediate dimensions, while optimization time grows more moderately owing to the ILP structure.

**ILP vs. Dynamic Programming for Bit Allocation.** As an efficient alternative solver to ILP, we evaluate a dynamic programming (DP) solver for the bit allocation problem. In our implementation, DP casts the row-wise assignment as a

*Table 17.* Time cost (in seconds) per transformer block for PRINMIX across model scales on a single L40 GPU.

| Model | Hidden Size | Interm. Size | Simulation | Optimization | Quantization | Total (s) |
|-------|-------------|--------------|------------|--------------|--------------|-----------|
| 7B    | 3584        | 18944        | 31.2       | 72.5         | 20.7         | 124.4     |
| 14B   | 5120        | 13824        | 33.4       | 128.6        | 26.0         | 188.0     |
| 70B   | 8192        | 28672        | 141.0      | 175.5        | 59.9         | 376.4     |

grouped knapsack problem: each row of $\mathbf{V}$ is one group, each candidate bit-width $j$ is an item with storage cost $B_j$ and row-wise error $\mathbb{E}^{\mathbf{V}}_{k,j}$, and exactly one item is selected from each group. Let $F(k, v)$ denote the minimum cumulative error after assigning one bit-width to each of the first $k$ groups under budget $v$. The transition updates are subject to $v \geq B_j$ and enough residual budget for the remaining groups:

$$F(k,v) = \min_{j}\{F(k-1, v - B_j) + \mathbb{E}^{\mathbf{V}}_{k,j}\} \tag{13}$$

Table 18 compares DP and ILP (SCIP) on Qwen2.5-Math-7B-Instruct. DP provides approximately $3\times$ optimization speedup and about $1.7\times$ end-to-end speedup, with only a 1.0 point drop in average score. In our current implementation, ILP retains two key advantages: (1) it handles the global bit-budget and active-bit-width constraints in Eq. (8) jointly, without additional outer-loop enumeration; and (2) it restricts the number of distinct bit-widths used in the mixed-precision schedule, which can simplify kernel implementation and improve inference efficiency in deployment.

*Table 18.* Comparison of ILP (SCIP) and DP bit allocators on Qwen2.5-Math-7B-Instruct. Time is reported per transformer block (in seconds).

| Allocator  | Simulation | Optimization | Quantization | Total (s) | Avg Score |
|------------|------------|--------------|--------------|-----------|-----------|
| DP         | 30.6       | 23.3         | 21.2         | 75.1      | 85.2      |
| SCIP (ILP) | 30.1       | 73.6         | 23.7         | 127.4     | 86.2      |

### C.6. Sensitivity to the Number of Candidate Bit-widths

In this section, we investigate the sensitivity of PRINMIX to the number of candidate bit-widths. We construct the candidates incrementally from $\{0, 2, 3, 4\}$ to $\{0, 2, 3, 4, 5, 6, 7, 8\}$ and quantize Qwen2.5-Math-7B-Instruct at $\alpha = 1/16$ in these 5 different configurations. As shown in Table 19, the performance gap between the best and worst average score is 0.6%, which demonstrates that PRINMIX is robust to the number of candidate bit-widths.

*Table 19.* Performance of PRINMIX using incrementally expanded candidate bit-widths.

| Candidate Bit-widths | Qwen2.5-Math-7B-Instruct | | |
|----------------------|---------|-------|------|
|                      | Math500 | GSM8K | AVG  |
| $\{0, 2, 3, 4\}$           | 76.7    | 94.2  | 85.5 |
| $\{0, 2, 3, 4, 5\}$        | 77.1    | 94.0  | 85.7 |
| $\{0, 2, 3, 4, 5, 6\}$     | 76.9    | 94.5  | 85.5 |
| $\{0, 2, 3, 4, 5, 6, 7\}$  | 77.3    | 94.4  | 85.9 |
| $\{0, 2, 3, 4, 5, 6, 7, 8\}$ | 77.7  | 94.6  | 86.2 |

### C.7. Inference Speed and Memory Cost

To demonstrate the impact of PRINMIX on inference speed and memory cost, we implement a simple Triton (Tillet et al., 2019) kernel for PRINMIX. We compare our kernel with naive aligned models. Since there is no packing function of Delta-CoMe, we use our packing function and kernel for the Delta-CoMe method.

Following the setup in Yao et al. (2024), we assess the end-to-end system performance under varying arrival rates and request distributions. We consider two types of model popularity distribution: 1) Uniform: all models are equally popular. 2) Skewed: model popularity follows a Zipf-$\alpha$ distribution. We evaluate the performance when serving 32 model variants of Qwen2.5-7B. Requests are sent to the serving system at a variable Poisson arrival rate ($\lambda$). To simplify, each request consists of 512 tokens, with the model generating one token as its response. We run the simulations for 100 seconds across different arrival rates and model distributions, measuring performance using two metrics: 1) end-to-end latency averaged over all requests; 2) Throughput, number of requests processed per second. All experiments are conducted on a single L40 GPU,

with 28G of memory for storing models and the remaining memory for inference.

As shown in the Table 20, PRINMIX improves the throughput 6x and decreases end-to-end 100x compared to the naive method, because rather than loading the whole full-precision parameters, PRINMIX quantizes the delta-parameters so that a GPU can load more delta-parameters and switch them easily between CPU and GPU.

*Table 20.* The Throughput and End-to-end system performance under varying arrival rates and request distributions when serving 32 model variants of Qwen2.5-7B.

|  | $\lambda = 0.5$ | | $\lambda = 1.0$ | |
| --- | --- | --- | --- | --- |
|  | Throughput(req/s) | E2E(s) | Throughput(req/s) | E2E(s) |
| Zipf ($\alpha = 1.5$) | | | | |
| Naive | 0.21 | 52.42 | 0.18 | 198.48 |
| Delta-CoMe | **0.42** | 0.55 | **0.87** | 0.68 |
| PRINMIX | **0.42** | **0.52** | **0.87** | **0.62** |
| Uniform | | | | |
| Naive | 0.07 | 253.93 | 0.08 | 481.42 |
| Delta-CoMe | **0.42** | 0.81 | **0.86** | 1.44 |
| PRINMIX | **0.42** | **0.79** | **0.86** | **1.17** |

## C.8. Analyzing the Bit Allocation Results

We investigate the bit allocation results across different weight types and layers using the Qwen2.5-Math-7B-Instruct model. Figure 4 shows the memory allocated for each bit-width. Overall, the bit allocation results for different weight types and layers are different. The V_Proj, K_Proj and O_proj in the self-attention layer exhibit a similar allocation trend. For the other four weight types, the bit allocation results differ. For instance, Down_Proj allocates more 2-bit units at the beginning compared to other weight types.

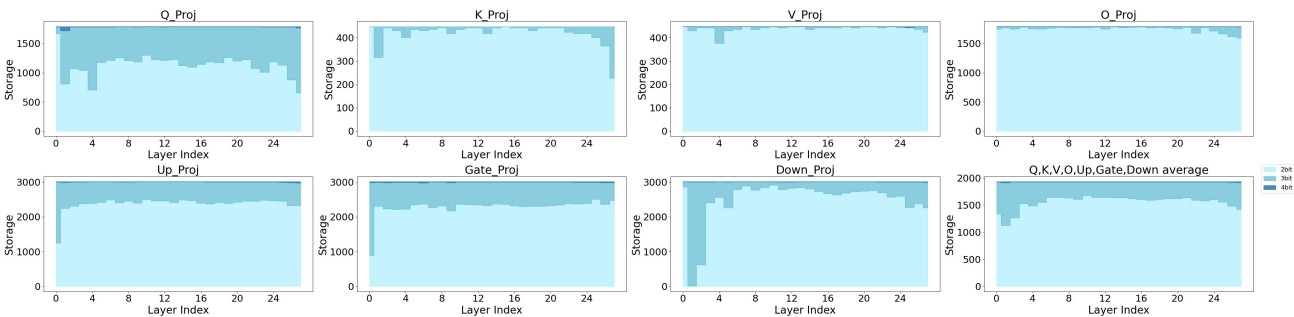

*Figure 4.* GPU memory usage with quantization bits across layers of Qwen2.5-Math-7B-Instruct.

Delta-CoMe (Ping et al., 2024) empirically posits that singular vectors corresponding to larger singular values are more significant and, therefore, necessitate higher-bit representations. We further examine whether PRINMIX adheres to this assumption, specifically by using singular values alone to evaluate importance. We compute the Kendall rank correlation coefficient $\tau$, between the bit sequence and the singular value sequence for each $\mathbf{W}$. The coefficient is a measure of rank correlation, ranging from -1 to 1, reflecting the similarity of the orderings of the data when ranked by each of the quantities. If the method strictly adhered to the assumption of using singular values alone for importance assessment, singular vectors with larger singular values would always receive higher bit-width, resulting in a consistent $\tau = 1$ across all $\mathbf{W}$. However, for the DeepSeek-R1-Distill-Qwen-7B model with PRINMIX, we observe a $\tau$ of 0.95 for the $\mathbf{W}$ of the key projection at layer 28. This indicates that PRINMIX goes beyond singular values, taking into account both the "scaling" term and the "difference" term.

## C.9. Analyzing the Quantization Error Across Weight Types and Layers

*Table 21.* Average quantization error ($\times$ 1e2) accross different type of linears with Eq. (1)."Low", "Mid", and "High" denote the first 9 layers, layers 9 to 17, and the last 10 layers, respectively. "All" and "Out" denote the average error across all activations and the average error of the top 1% of activations.

| Param | Q_proj | | | | | | Param | K_proj | | | | | |
|---|---|---|---|---|---|---|---|---|---|---|---|---|---|
| Layer | Low | | Mid | | High | | Layer | Low | | Mid | | High | |
| Type | All | Out | All | Out | All | Out | Type | All | Out | All | Out | All | Out |
| Low-Rank | 0.26 | 0.32 | 0.54 | 0.76 | 1.33 | 1.64 | Low-Rank | 0.06 | 0.07 | 0.11 | 0.13 | 0.19 | 0.29 |
| BitDelta | 0.18 | 0.37 | 0.27 | 0.37 | 0.68 | 1.00 | BitDelta | **0.03** | **0.03** | **0.05** | **0.06** | **0.08** | **0.12** |
| Delta-CoMe | 0.13 | 0.14 | 0.32 | 0.41 | 0.81 | 0.91 | Delta-CoMe | **0.03** | **0.03** | 0.06 | 0.07 | 0.12 | 0.21 |
| PRINMIX | **0.10** | **0.11** | **0.25** | **0.32** | **0.64** | **0.73** | PRINMIX | **0.03** | **0.03** | **0.05** | 0.07 | 0.10 | 0.18 |

| Param | V_proj | | | | | | Param | O_proj | | | | | |
|---|---|---|---|---|---|---|---|---|---|---|---|---|---|
| Layer | Low | | Mid | | High | | Layer | Low | | Mid | | High | |
| Type | All | Out | All | Out | All | Out | Type | All | Out | All | Out | All | Out |
| Low-Rank | 0.03 | 0.03 | 0.06 | 0.08 | 0.39 | 1.11 | Low-Rank | 0.23 | 0.40 | 0.70 | 1.54 | 8.52 | 69.00 |
| BitDelta | **0.01** | **0.01** | **0.03** | **0.03** | **0.18** | 0.69 | BitDelta | 0.10 | 0.14 | **0.28** | 0.46 | 10.44 | 895.98 |
| Delta-CoMe | 0.02 | 0.02 | 0.04 | 0.05 | 0.24 | 0.85 | Delta-CoMe | 0.08 | 0.13 | 0.32 | 0.47 | 3.53 | 17.02 |
| PRINMIX | 0.02 | 0.02 | 0.04 | 0.05 | 0.21 | 0.67 | PRINMIX | **0.07** | **0.12** | 0.30 | **0.45** | **3.18** | **22.31** |

| Param | Up_proj | | | | | | Param | Gate_proj | | | | | |
|---|---|---|---|---|---|---|---|---|---|---|---|---|---|
| Layer | Low | | Mid | | High | | Layer | Low | | Mid | | High | |
| Type | All | Out | All | Out | All | Out | Type | All | Out | All | Out | All | Out |
| Low-Rank | 4.78 | 4.50 | 2.67 | 3.18 | 13.70 | 14.95 | Low-Rank | 6.35 | 3.85 | 3.16 | 0.72 | 13.53 | 4.02 |
| BitDelta | 4.71 | 3.85 | **1.19** | **1.32** | 13.30 | 11.61 | BitDelta | 9.01 | 4.47 | 1.60 | 0.65 | 10.32 | 5.87 |
| Delta-CoMe | 2.10 | 2.08 | 1.60 | 1.90 | 7.67 | 9.37 | Delta-CoMe | 2.64 | 2.90 | 1.88 | 0.84 | 7.73 | 3.02 |
| PRINMIX | **1.83** | **1.74** | 1.36 | 1.59 | **6.58** | **8.89** | PRINMIX | **2.28** | **2.22** | 1.57 | **0.59** | 6.65 | **2.07** |

| Param | Down_proj | | | | | | Param | AVG | | | | | |
|---|---|---|---|---|---|---|---|---|---|---|---|---|---|
| Layer | Low | | Mid | | High | | Layer | Low | | Mid | | High | |
| Type | All | Out | All | Out | All | Out | Type | All | Out | All | Out | All | Out |
| Low-Rank | 1.05 | 5.52 | 3.28 | 4.94 | 110.20 | 7470.34 | Low-Rank | 1.82 | 3.67 | 1.50 | 2.84 | 21.12 | 1890.34 |
| BitDelta | 1.21 | 2.35 | 0.87 | 1.45 | 115.60 | 11735.05 | BitDelta | 2.18 | 2.81 | 0.61 | 1.08 | 21.51 | 3162.58 |
| Delta-CoMe | 0.33 | 1.86 | 1.05 | 1.57 | 32.66 | 1851.91 | Delta-CoMe | 0.76 | 1.79 | 0.75 | 1.33 | 7.54 | 470.82 |
| PRINMIX | **0.31** | **1.62** | **1.02** | **1.43** | **30.30** | **1669.95** | PRINMIX | **0.66** | **1.46** | **0.66** | **1.12** | **6.81** | **426.20** |

