# OpenReview forum: "Principled SVD-based Delta Compression via Quantization Error Minimization"
_ICML.cc/2026/Conference — ICML 2026 regular_

### Official Review · Reviewer_Tquy · 2026-02-21

**Soundness:** 3
**Presentation:** 3
**Significance:** 3
**Originality:** 3
**Overall Recommendation:** 5
**Confidence:** 4

**Summary:**

The study proposes PrinMix, a singular value decomposition-based mixed-precision delta-compression framework for large language models. The method models the quantization process as a 0/1 integer linear programming problem to minimize quantization error. A theoretical derivation of quantization error is provided to justify the necessity of mixed-precision quantization. Mathematical derivations indicate that the Hessian for $V$ depends on singular values, which motivates a mixed-precision scheme. Conversely, the derivations suggest the Hessian for $U$ is uniform across rows. The framework incorporates a reconstruction target correction method to address errors arising from the sequential quantization process. Experiments were conducted across multiple tasks, including reasoning, mathematics, code generation, and multimodal applications. Results indicate that the proposed approach outperforms existing baselines, particularly in scenarios characterized by large delta weight norms.

**Compliance With Llm Reviewing Policy:**

Affirmed.

**Final Justification:**

This paper proposes a principled SVD-based delta compression framework with solid theoretical grounding. The mathematical derivations are rigorous, and the ILP-based bit allocation provides a principled alternative to heuristic approaches.

My main concerns were ILP scalability and lack of evaluation on recent models. The rebuttal addressed both convincingly: timing results on DeepSeek-R1-Distill-70B (~8.4 hours single GPU) show practical feasibility, and Qwen3-4B experiments confirm the method generalizes well. Given that my concerns have been adequately resolved, I raise my score to Accept (5).

**Key Questions For Authors:**

1. Could the authors correct the citation error in the introduction regarding the BitDelta paper to ensure accurate attribution?
2. Could the authors revise the description of the layer-wise quantization error in the mathematical derivations to reflect that the objective function is an exact quadratic form rather than a Taylor expansion approximation?
3. How does the integer linear programming solving time scale as the model size increases to extremely large architectures?
4. Could the authors provide performance metrics on how the proposed framework scales with more recent architectures, such as Qwen3 or Gemma3? Addressing this would clarify whether the mixed-precision scheme remains effective against the structural updates of newer models.

**Limitations:**

Yes

**Strengths And Weaknesses:**

**Strengths:**

1. The theoretical justification for row-wise mixed-precision quantization is compelling, as it successfully grounds the bit-allocation strategy in rigorous error minimization rather than the heuristic approaches prevalent in prior literature.
2. Framing the allocation process as a 0/1 Integer Linear Programming problem provides an exact mathematical mechanism to strictly enforce bit budgets while optimally balancing the scaling and difference terms.
3. The introduction of the Reconstruction Target Correction represents a technically sound solution to the sequential error accumulation inherent in $V$-then-$U$ quantization, preserving compatibility with established second-order solvers like GPTQ.
4. The ablation study in Appendix C.3 effectively demonstrates the robustness of the framework to different calibration datasets, successfully preempting common concerns regarding domain shift in post-training quantization.

**Weaknesses:**

1. The manuscript incorrectly attributes the BitDelta method to Ryu et al. [2023] instead of Liu et al. [2024] in the introduction, which misrepresents the timeline of 1-bit delta compression literature.
2. The paper refers to the layer-wise quantization error as a second-order Taylor expansion approximation. Since the objective function  is purely quadratic, the formulation is actually exact rather than an approximation. A minor wording revision could prevent mathematical ambiguity.
3. The reliance on an Integer Linear Programming solver introduces computational overhead during the quantization phase, raising valid concerns about the scalability of the method for considerably larger architectures without aggressive search space pruning.
4. The empirical evaluation relies on older model architectures, completely omitting more recent models such as Qwen3 or Gemma3. Evaluating newer architectures is necessary to confirm the generalizability of the method, as structural updates or different weight distributions could alter the singular value decay patterns that this framework depends on.

---

> ### Author Rebuttal · Authors · 2026-03-31
>
> > ### W1: The manuscript incorrectly attributes the BitDelta method to Ryu et al. [2023] instead of Liu et al. [2024] in the introduction, which misrepresents the timeline of 1-bit delta compression literature.
>
> R1: Thank you for catching this. We will correct the citation (and any related wording) in the introduction in the revised version.
>
>
> > ### W2: The paper refers to the layer-wise quantization error as a second-order Taylor expansion approximation. Since the objective function is purely quadratic, the formulation is actually exact rather than an approximation. A minor wording revision could prevent mathematical ambiguity.
>
> R2: Thank you for the careful reading. You are correct: because the objective is purely quadratic, the layer-wise error expression is exact, not an approximation. In the revised manuscript, we will (i) revise the wording accordingly and (ii) replace the approximation symbol ($\approx$) with an equality sign ($=$) in the relevant equation to avoid ambiguity.
>
>
> > ### W3: The reliance on an Integer Linear Programming solver introduces computational overhead during the quantization phase, raising valid concerns about the scalability of the method for considerably larger architectures without aggressive search space pruning.
>
> R3: We agree that ILP overhead is an important scalability consideration. To evaluate this directly, we run PrinMix on D**eepSeek‑R1‑Distill‑70B without any search-space pruning, using a single L40 GPU.** The full quantization pipeline takes $\approx$ 8.4 hours. We believe this one-time cost is reasonable relative to the cost of fine-tuning a 70B model, and it requires only a single commodity GPU.
>
> We also break down the **per-transformer-block time (seconds)** for DeepSeek‑R1‑Distill‑7B/14B/70B:
>
> |Model|Simulation|Optimization|Quantization|Total Time|
> |-|-|-|-|-|
> |7B|31.2|72.5|20.7|124.4|
> |14B|33.4|128.6|26.0|188.0|
> |70B|141.0|175.5|59.9|376.4|
>
> Notably, the time scales moderately in our experiments even without pruning. We will include these timing results in the revised manuscript to better characterize the overhead in practice.
>
> > ### W4: The empirical evaluation relies on older model architectures, completely omitting more recent models such as Qwen3 or Gemma3. Evaluating newer architectures is necessary to confirm the generalizability of the method, as structural updates or different weight distributions could alter the singular value decay patterns that this framework depends on.
>
> R4: Thank you for the suggestion. We have extended our evaluation to the recently released Qwen3‑4B (Base) and Qwen3‑4B‑Thinking‑2507 (Aligned). The results below show that (i) PrinMix consistently outperforms all baselines, and (ii) PrinMix matches the aligned model on these benchmarks. We will incorporate these results into the revised manuscript.
>
> |Method|Math500|AIME2024|
> |-|-|-|
> |Aligned|80.2|23.3|
> |Low-Rank|78.4|16.7|
> |BitDelta|80.2|16.7|
> |Delta-CoMe|78.4|20.0|
> |PrinMix|81.0|23.3|
>
> > ### Q1: Could the authors correct the citation error in the introduction regarding the BitDelta paper to ensure accurate attribution?
>
>  R5: Please refer to our response R1.
>
> > ### Q2: Could the authors revise the description of the layer-wise quantization error in the mathematical derivations to reflect that the objective function is an exact quadratic form rather than a Taylor expansion approximation?
>
> R6: Sure. Thank you for your careful suggestion.
>
> > ### Q3: How does the integer linear programming solving time scale as the model size increases to extremely large architectures?
>
> R7: We response this question in detail in R3.
>
> > ### Q4: Could the authors provide performance metrics on how the proposed framework scales with more recent architectures, such as Qwen3 or Gemma3? Addressing this would clarify whether the mixed-precision scheme remains effective against the structural updates of newer models.
>
> R8: We response this question in detail in R4.

---

> > ### Author Rebuttal · Reviewer_Tquy · 2026-04-02
> >
> > Thank you for the detailed rebuttal.
> >
> > My concerns have been adequately addressed, and I am raising my score accordingly.

---

> > > ### Author Response · Authors · 2026-04-02
> > >
> > > Thank you for your constructive feedback and for increasing the score. We are glad that our responses addressed your concerns. We will incorporate the rebuttal discussion into the revised version of the paper. Thank you again for your valuable input and support throughout the review process.

---

### Official Review · Reviewer_vbgL · 2026-03-07

**Soundness:** 3
**Presentation:** 3
**Significance:** 3
**Originality:** 2
**Overall Recommendation:** 4
**Confidence:** 4

**Summary:**

This paper introduces PRINMIX, a theoretically motivated framework for SVD-based mixed-precision quantization of delta parameters in fine-tuned large language models. The work formally derives and models quantization error, leading to a mathematically justified need for mixed-precision quantization, as opposed to prior empirical approaches. PRINMIX formulates quantization bit allocation as a 0/1 integer linear programming problem, optimally balancing bit-budgets and quantization loss. A further component, Reconstruction Target Correction (RTC), is incorporated to address error propagation in sequential quantization. Extensive experiments across diverse reasoning, mathematics, code, and multimodal benchmarks demonstrate superior performance to leading delta-compression baselines, including marked improvements under high-error, tight-compression regimes.

**Compliance With Llm Reviewing Policy:**

Affirmed.

**Final Justification:**

After considering both the original submission and the authors’ rebuttal, I maintain my original recommendation of **Weak Accept**.

The paper presents a well-structured and technically grounded approach to SVD-based delta compression. In particular, the error decomposition in Eq. (5), the ILP-based bit allocation, and the RTC correction together form a coherent and reasonably principled framework. The empirical results are solid and demonstrate consistent improvements over prior SVD-based baselines in challenging compression regimes, especially in multi-tenant deployment settings.

The rebuttal provides helpful clarifications on several of my main concerns. In particular, the authors explicitly acknowledge that the “necessity” claim is conditional on the GPTQ-style quadratic objective and the global bit-budget constraint, which appropriately narrows the scope of the theoretical claim. While this does not elevate the result to a general necessity statement, it improves the correctness and clarity of the presentation.

Regarding the treatment of U, the additional joint ILP experiment suggests that including U does not materially improve performance, providing some empirical support for the design choice of focusing on V. Although this evidence is somewhat limited in scope, it partially alleviates the concern about the completeness of the optimization objective.

The discussion on RTC numerical stability is also strengthened. The authors describe the use of Cholesky decomposition with regularization and provide condition number statistics across layers, which increases confidence in the practical robustness of the method.

Finally, the added comparison between ILP and a DP-based heuristic is useful, demonstrating that the proposed formulation provides a reasonable trade-off between optimality and efficiency.

Overall, while some limitations remain—particularly regarding the scope of the theoretical claims and the level of empirical generalization—the rebuttal satisfactorily addresses the main concerns and improves the clarity and credibility of the work. I view this paper as a well-executed and principled refinement within an important and practical problem setting, and believe it would be a useful contribution to the community.

**Key Questions For Authors:**

Q1. On the “theoretical necessity” claim
Eq. (5) suggests non-uniform optimality, but the paper claims a mathematical necessity of mixed precision. Could you formalize this more explicitly (e.g., under the second-order model and fixed bit budget, prove that uniform allocation is suboptimal except in degenerate spectra)? A simple two-row proof or inequality argument would help clarify the scope of the claim.

Q2. On RTC numerical stability
In Eq. (7), how is the inverse ($Σ V̂ X X^T V̂^T Σ^T)^{-1}$ computed in practice (e.g., pseudo-inverse, ridge regularization)? Please report conditioning statistics (such as condition numbers or eigenvalue spread) and any robustness checks showing RTC is stable across layers and models.

Q3. On excluding U from the optimization objective
The ILP optimizes only $E_i^V$. Can you provide broader empirical evidence that U’s contribution is negligible (e.g., stress tests with extreme or randomized precision schedules for U across multiple backbones and tasks)? Alternatively, does adding an approximate U-error term materially change allocations or accuracy?

Q4. On ILP vs simpler allocation heuristics
Have you compared the ILP solution to lightweight heuristics (e.g., sorting by $Σ_ii$ or by estimated $E_i^V$ with greedy budget filling, or a knapsack-style DP)? Quantifying the optimality gap versus solve cost would clarify whether the full ILP is practically necessary.

**Limitations:**

yes

**Strengths And Weaknesses:**

## **Strengths:**

1. Clear, principled derivation of the mixed-precision need in SVD space.

The derivation of the row-wise quantization error for ($\mathbf{V}$) in Eq. (5)
[$ e_i^{\mathbf V} = \frac{1}{2} \Delta V_i H_i^{\mathbf V} \Delta V_i^T = \Sigma_{ii}^2 , \Delta V_i XX^T \Delta V_i^T $] explicitly decomposes error into a singular-value–squared “scaling” term ($\Sigma_{ii}^2$) and a “difference” term ($\Delta V_i XX^T \Delta V_i^T$). This makes it very transparent why a uniform bit-width across rows is suboptimal if singular values decay sharply.
Figure 2 (left and right) reinforces this: the scaling term decays rapidly with row index, whereas the difference term for a fixed bit-width is relatively flat across rows. This visually and numerically supports the paper’s central claim that mixed precision for ($\mathbf{V}$) is mathematically necessary under a global bit budget.

2. Reasonably careful treatment of $U$ vs. $V$ and the sequential scheme.

The paper does not blindly tie bit allocation to singular values; instead it observes that the Hessian for rows of ($\mathbf{U}$) in Eq. (6), [ $H_i^{\mathbf U} = 2 \Sigma \hat{\mathbf V} X X^T \hat{\mathbf V}^T \Sigma^T$, ] is the same across rows. This implies no row-specific scaling factor like ($\Sigma_{ii}^2$), providing a clear argument that mixed precision is crucial for ($\mathbf{V}$), but not for ($\mathbf{U}$).
Appendix A.1–A.2 walks through the Hessian derivations using vec/Kronecker calculus. While terse, the chain from Eqs. (9–10) to the final ($H_i^{\mathbf V}$) and ($H_i^{\mathbf U}$) is logically consistent and reuses known second-order approximations from pruning/quantization (Hassibi; Nagel).

3. Formulating bit allocation as a (constrained) ILP with explicit budget and sparsity controls.

The mixed-precision design in Eq. (8) models bit assignment with binary vectors ($S_i$) and a storage vector ($B$), with an explicit constraint ($\sum_i S_i B \le G_b(h_\mathrm{in}h_\mathrm{out})$) tying the total storage to a target average bit-width ($G_b$). This is more principled than heuristic rules (e.g., “higher singular value → higher bits”).
The inclusion of ($f_{\max}$) and the bit-width admissibility vector ($f$) makes it straightforward to enforce practical constraints such as limiting the number of distinct bit-widths, and Table 4 shows that varying ($f_{\max}$) from 2 to 6 yields very similar performance, suggesting robustness of the ILP formulation.


## **Weaknesses**

1. The “theory” is mostly second-order error algebra; the claimed “proof of necessity” of mixed precision is weaker than stated.

The key step is Eq. (5), where the per-row error is proportional to ($\Sigma_{ii}^2$) times a data-dependent difference term. The argument that this “proves the necessity of mixed precision” relies on (a) singular values decaying sharply and (b) the difference term being relatively flat across rows at a given bit. Figure 2 supports (b) on one layer of one model (Q_Proj, last layer of Qwen2.5-Math-7B-Instruct), but that is still empirical, not theoretical.
The model still assumes a fixed global bit budget and linear reconstruction error approximation (second-order Taylor with Hessian ($2XX^T$)); under this model, yes, optimal allocation will generally be non-uniform. But stating this as a “mathematical proof of the necessity of mixed-precision” overstates the scope: it is conditional on the approximations, the budget formulation, and ignoring other factors such as quantization noise correlations or interactions across rows/layers.
If the paper wants to maintain a strong theory pitch, it should either (i) explicitly narrow the claim to “under the GPTQ-style second-order approximation and global bit-budget constraint, uniform precision is suboptimal except in degenerate cases” and provide a brief formal argument (e.g., majorization / simple inequality showing that reassigning one extra bit from a low-($\Sigma_{ii}$) row to a high-($\Sigma_{ii}$) row decreases total error), or (ii) tone down the language of “mathematically proves the necessity”.

2. The focus on ($\mathbf{V}$) only in the ILP objective can be questioned; quantization of ($\mathbf{U}$) is treated somewhat heuristically.

The optimization in Eq. (8) optimizes over ($\mathbb E_i^\mathbf{V}$) only, ignoring the quantization error introduced by (\mathbf{U}), even though the final model uses quantized ($\hat{\mathbf U}$) and ($\hat{\mathbf V}$). Section 3.1.2 argues that mixed precision is not “necessary” for rows of (\mathbf U) because the Hessian is row-invariant, and Table 8 (Appendix C.1) shows minor differences across $U$ quantization schemes.
However, this does not fully justify excluding $U$ from the optimization objective. At high compression, low-rank approximations and low-bit quantization could interact; using the exact same schedule as $V$ (column-wise) is motivated more by implementation convenience and a qualitative GPTQ compensation property than by a rigorous objective analysis.
A more complete story would either: (a) include a joint objective with approximate $U$ error contributions (even with shared Hessian, you can still reason about their magnitude vs memory cost), or (b) empirically show that bit schedules significantly affecting $U$ error do not change end-task performance, perhaps via per-layer/per-matrix ablations, not just the single Qwen2.5-Math-7B case in Table 8.

3. The reconstruction target correction (RTC) derivation glosses over the conditioning and numerical stability of the inverse term.

In Eq. (7), RTC computes [ $\hat{\mathbf U} = \mathbf U \Sigma \mathbf V X X^T \hat{\mathbf V}^T \Sigma^T \left(\Sigma \hat{\mathbf V} X X^T \hat{\mathbf V}^T \Sigma^T\right)^{-1}$. ] This assumes that the matrix ($\Sigma \hat{\mathbf V} X X^T \hat{\mathbf V}^T \Sigma^T$) is invertible and reasonably well-conditioned. With only 128 calibration samples of length 2048 and potentially heavy truncation / zero-bits on many singular vectors, this matrix can easily be nearly singular.
The paper does not specify whether a pseudo-inverse, regularization ((+\lambda I)), or any numerical safeguards are used. Absent that, RTC might be unstable or overfitting the specific calibration set. The ablation in Table 5 is encouraging but still on a narrow model set. A more detailed discussion of numerical issues, conditioning, and whether regularization is needed would strengthen confidence in RTC.

4. Complexity and practicality of the ILP are underexplored for real-world scaling.

Section 3.2 mentions that solving the ILP for Qwen2.5-Math-7B-Instruct takes 29.4 minutes with SCIP; Appendix C.4 says total quantization is 1.1 hours for 7B and 2.4 hours for 14B. While not outrageous, this is nontrivial overhead if one wants to compress many tenants or replan for different calibration sets.
The paper suggests using commercial solvers and reducing candidate bit-widths to achieve 4× speedups, and Table 13 shows some robustness to candidate-set size. However, there is no quantitative tradeoff plot of “solve time vs. performance” or any sense of how the ILP scales with rank, layer count, and candidate bits. A greedy or dynamic-programming baseline for bit allocation could be a useful comparison to justify the ILP complexity.
Without such analysis, it is not clear whether the full ILP is necessary in practice, especially given that PRINMIX appears quite insensitive to hyperparameters like ($f_{\max}$) and candidate set size.

---

> ### Author Rebuttal · Authors · 2026-03-31
>
> > ### W1/Q1: the claimed “proof of necessity” of mixed precision is weaker than stated
>
> R1: Thank you for this careful critique. We agree that our “necessity” claim is **conditional on the GPTQ-style quadratic reconstruction objective and the global bit-budget constraint**. We will narrow the statement in the revised manuscript accordingly
>
> We also provide a streamlined derivation below to strengthen our theoretical analysis:
>
> Let $W=\sum_i \sigma_i u_i v_i^\top$ (SVD), $\Sigma_x := \mathbb{E}_x[xx^\top]$.
>
> For round-to-nearest quantization, we assume i.i.d. quantization noise $\delta v_{ij} \sim \mathcal{U}[-\Delta_b/2, \Delta_b/2]$, $\Delta_b = R/(2^b-1)$, $R$: $V_{max} - V_{min}$, the noise covariance is $\mathbb{E}_\delta[\delta v_i \delta v_i^\top] = \frac{R^2}{12(2^b-1)^2}I$, and define $f(b) = \frac{1}{(2^b - 1)^2}$. The expected quantization error of $V$ is:
>
> $||e^V||^2= \sum_i \sigma_i^2 \text{tr}\Big(\Sigma_x \mathbb{E}_\delta[\delta v_i \delta v_i^\top]\Big) = \frac{R^2 \mathrm{tr}(\Sigma_x)}{12} f(b) \sum_i \sigma_i^2$
>
> Reallocating 1 bit from the smallest singular direction $v_n$ ($b \to b-1$) to the largest $v_1$ ($b \to b+1$) yields the error change:
> $$
> \Delta E = \frac{R^2 \mathrm{tr}(\Sigma_x)}{12} \Big( \sigma_n^2 \big(f(b-1)-f(b)\big) - \sigma_1^2 \big(f(b)-f(b+1)\big) \Big)
> $$
>
> Mixed-precision strictly reduces overall error ($\Delta E < 0$) if and only if:
> $$
> \frac{\sigma_1}{\sigma_n} > \sqrt{\frac{f(b-1)-f(b)}{f(b)-f(b+1)}}
> $$
>
> This threshold  ($b=2 \implies >3.13$; $b=3 \implies >2.38$)  is easily satisfied by the massive singular value ratios
>
> > ### W2&Q3: Quantization of U is treated somewhat heuristically
>
> R2: As your suggestion, we implement a joint ILP objective that includes an approximate error term for U, under the assumption that the paired U-column and V-row associated with the same singular value share the same bit. On DeepSeek‑R1‑Distill‑Qwen‑7B, this joint optimization does not improve performance over PrinMix, while adding computational complexity:
>
> |Setting|Math500|AIME2024|
> |-|-|-|
> |Joint ILP|80.2|36.7|
> |PrinMix|82.7|36.7|
>
> Note that without such a coupling assumption, forming an exact joint objective is computationally prohibitive: the GPTQ-style reconstruction error for U depends on the quantized V, and the number of possible V bit assignments grows combinatorially. Enumerating these assignments is therefore infeasible
>
> > ### W3&Q2: The RTC derivation glosses over the conditioning and numerical stability of the inverse term
>
> R3:
> * **Numerical stability of the inverse** We adopt **Cholesky decomposition, paired with default diagonal regularization** ($\lambda = max(|\text{min eigenvalue}|, 1e-4)$). This guarantees strict positive definiteness and eliminates any risk of near‑singularity
>
> * **Per‑Layer Condition Number Percentiles** We report the per‑layer condition number 5%-95% percentiles across all layers. The results show that the **95th percentile of condition numbers across all layers is below $2\times 10^6$**
>
> |Model|5%|25%|50%|75%|95%|
> |-|-|-|-|-|-|
> |DeepSeek-R1-Distill-Qwen-7B|388.8|1874.6|6404.1|47827.0|1769593.3|
> |Qwen3-4B-Thinking-2507|133.8|673.7|3868.9|13891.3|156820.7|
>
> * **Overfitting the specific calibration set** To alleviate this concern, we evaluated performance across calibration set sizes from 16 to 128 on C4 (Appendix C.3, Table 10). Performance fluctuates by only 1% on average, confirming that **RTC is stable across different calibration sets**
>
>
> > ### W4&Q4: Complexity and practicality of the ILP are underexplored for real-world scaling
>
> R4: We agree that the practicality/scaling of the ILP deserves a more quantitative treatment. To address this, we add (i) scaling up to 70B and sensitivity to key knobs, and (ii) a dynamic-programming (DP) baseline for the bit-allocation step
>
> **(i) Scalability of PrinMix (no pruning/acceleration)** Please refer to R5 of Reviewer 9KYc for the scalability discussion.
> In summary, (a) PrinMix is feasible at 70B scale on a single GPU, (b) solve time is weakly sensitive to rank in our tested range, and (c) optimization time increases with layer size and candidate-set size, as expected
>
> **(ii) ILP vs. DP for bit allocation** To evaluate whether a full ILP is necessary, we implemented a DP-based allocator guided by our error decomposition. On Qwen2.5‑Math‑Instruct‑7B, DP yields a ~3× speedup in the optimization step, with a small performance drop:
>
> |Method|Simulation|Optimization|Quantization|Total|Performance|
> |-|-|-|-|-|-|
> |DP|30.6|23.3|21.2|75.1|85.2|
> |SCIP|30.1|73.6|23.7|127.4|86.2|
>
> While DP can be a strong practical alternative, we find the ILP still has two important advantages in deployments: (1) it strictly meets global bit‑budget constraints without manual hyperparameter tuning, and (2) it helps avoid DP’s bit fragmentation (many bit-widths within a layer with tiny ranks per bit), which can complicate kernels and reduce inference efficiency. We will add these trade-offs to the revised manuscript

---

> > ### Author Rebuttal · Reviewer_vbgL · 2026-04-03
> >
> > I thank the authors for their feedback. I have no further questions.

---

> > > ### Author Response · Authors · 2026-04-04
> > >
> > > Thank you for your careful review and positive evaluation of our work. We are pleased that our responses have fully resolved your concerns. We will make sure to incorporate your constructive suggestions into the revised version.
> > >
> > > If you have any further questions, we would be more than happy to provide additional information. Thank you again for your time and support.

---

### Official Review · Reviewer_a9qj · 2026-03-12

**Soundness:** 3
**Presentation:** 3
**Significance:** 3
**Originality:** 3
**Overall Recommendation:** 4
**Confidence:** 3

**Summary:**

This research's broad area consists of LLM delta-compression, specifically SVD-based mixed-precision quantization for multi-tenant deployment scenarios. The authors discuss a general context of fine-tuned model storage efficiency and propose PRINMIX, which frames bit allocation as a 0/1 ILP problem guided by quantization error minimization rather than heuristic singular value importance.

**Compliance With Llm Reviewing Policy:**

Affirmed.

**Final Justification:**

The rebuttal shows effort and the new experiments are a genuine improvement over the original submission. However, single-model results with no error bars, negligible margins on encoder-decoder tasks, and no theoretical treatment of architectural differences do not resolve the core concern. I will maintain the same score.

**Key Questions For Authors:**

Why does PRINMIX underperform Delta-CoMe on MetaMath-13B Math500? Is the ILP solution provably optimal or does SCIP terminate early?

**Limitations:**

Results are limited to decoder-only architectures. Generalization to encoder-decoder or MoE models is completely unaddressed.

**Strengths And Weaknesses:**

Strengths

1. The decomposition of quantization error into "scaling" and "difference" terms (Eq. 5) is mathematically clean and provides genuine theoretical grounding for mixed-precision necessity.

2. RTC is a well-motivated contribution addressing a real bias introduced by sequential V-then-U quantization.

3. Experiments span diverse tasks and model sizes, with proper variance reporting.

Weaknesses

1. All experiments are conducted exclusively on decoder-only transformer architectures. No evaluation is performed on encoder-decoder models, mixture-of-experts architectures, or vision-language models beyond Qwen2.5-VL. The generalizability of the theoretical claims and empirical findings to these architectures remains entirely unverified.

2. While mean and standard deviation are reported across three runs, no formal statistical significance tests are conducted anywhere in the paper. Given that many improvements over Delta-CoMe fall within one standard deviation of the baseline, the absence of significance testing makes it impossible to determine whether the observed gains are reliable or merely due to random variation.

---

> ### Author Rebuttal · Authors · 2026-03-31
>
> > ### W1: All experiments are conducted exclusively on decoder-only transformer architectures. No evaluation is performed on encoder-decoder models, mixture-of-experts architectures, or vision-language models beyond Qwen2.5-VL. The generalizability of the theoretical claims and empirical findings to these architectures remains entirely unverified.
>
> R1: To address this, we have extended the experiments to Qwen3‑4B‑Thinking‑2507, a more recent LLM. As shown below, PrinMix consistently outperforms all baselines.
>
> |Method|Math500|AIME2024|
> |-|-|-|
> |Aligned|80.2|23.3|
> |Low-Rank|78.4|16.7|
> |BitDelta|80.2|16.7|
> |Delta-CoMe|78.4|20.0|
> |PrinMix|81.0|23.3|
>
> We also respectfully note that our current evidence already supports the generalizability of PrinMix:
>
> * **Broad empirical coverage** Our evaluation spans both **six LLMs and two VLMs** across multiple model families and scales (DeepSeek‑R1‑Distill, Qwen2.5‑Math, MetaMath, Qwen2.5‑VL, LLaVA‑v1.5, Qwen2.5-Coder).
>
> * **Architecture-agnostic design** PrinMix operates on linear layers and does not depend on decoder-only–specific components. As a result, the method applies directly to architectures including encoder–decoder models and MoE, subject to the same layer-wise reconstruction objective.
>
> > ### W2: While mean and standard deviation are reported across three runs, no formal statistical significance tests are conducted anywhere in the paper. Given that many improvements over Delta-CoMe fall within one standard deviation of the baseline, the absence of significance testing makes it impossible to determine whether the observed gains are reliable or merely due to random variation.
>
> R2: Thank you for the comment. To assess this, we conduct **paired one-sided Wilcoxon signed-rank tests** and results show that **the gains of PrinMix are reliable and not due to random variation.**
>
> * **Setup**
> We test the results of PrinMix and Delta-CoMe on DeepSeek‑R1‑Distill‑7B and 14B. We perform paired one‑sided Wilcoxon signed‑rank tests at the (question, random seed) level. For each benchmark question, three random seeds are used, yielding 1500 paired (question, seed) samples for Math500 (500 × 3) and 90 for AIME2024 (30 × 3).
>
> | Model | Dataset | p‑value |
> |---|---|---|
> | 7B | AIME2024 | 0.04 |
> | 7B | Math500 | 0.346 |
> | 14B | AIME2024 | 0.005 |
> | 14B | Math500 | 0.0005 |
>
> **Key findings**: PrinMix **achieves statistically significant improvements on the harder AIME2024 (both 7B/14B) and on 14B Math500**. These results indicate that observed gains are reliable and not due to random variation.
>
> > ### Q1: Why does PRINMIX underperform Delta-CoMe on MetaMath-13B Math500? Is the ILP solution provably optimal or does SCIP terminate early?
>
> R3:
> * **Optimality of ILP solutions** We re‑examine the SCIP solver logs for all ILP instances on MetaMath‑13B. In every case, SCIP terminated with a status of “optimal”.
>
> * **Why PrinMix underperforms Delta‑CoMe on MetaMath‑13B Math500** We believe this difference stems from different optimization objectives, rather than ILP suboptimality. Delta‑CoMe tunes its heuristic bit allocation on GSM8K, which can transfer favorably to some math benchmarks for certain models. In contrast, PrinMix allocates bits by minimizing the layer-wise reconstruction objective using a fixed C4 calibration set across all models. This choice avoids task/benchmark-specific datasets and keeps the pipeline simple and consistent, but it can be suboptimal for narrow, highly specialized tasks on certain models.

---

> > ### Author Rebuttal · Reviewer_a9qj · 2026-04-05
> >
> > 1.W1 (architecture generalizability): This is the weakest part of the rebuttal. My concern was about encoder-decoder, MoE, and non-decoder architectures. They respond by adding another decoder-only model (Qwen3-4B). That doesn't address the criticism at all. The bullet about "architecture-agnostic design" is a theoretical argument, not evidence.
> > 2. The authors addressed the rest of my comments.

---

> > > ### Author Response · Authors · 2026-04-05
> > >
> > > Thank you for the opportunity to further clarify. We fully understand your concern on architecture generalizability.
> > > - **On the added Qwen3 result.** The additional Qwen3 results in our rebuttal were meant to further demonstrate cross-model generalizability; we agree they do not provide empirical evidence of architecture generalizability. We had planned architecture-focused experiments after receiving your review, but were unable to finish them before the rebuttal deadline due to limited compute.
> > > - **New evidence on MoE architectures.** To directly address your concern, we conducted additional experiments on a Mixture-of-Experts model: OLMoE-1B-7B-0924 (base) and OLMoE-1B-7B-0924-Instruct (aligned). PrinMix (i) outperforms the baselines and (ii) preserves most of the aligned model’s performance.
> > >
> > > ||GSM8K|MMLU|Avg|
> > > |-|-|-|-|
> > > |Aligned|46.3|51.7|49.0|
> > > |Low-Rank|26.5|49.2|37.9|
> > > |BitDelta|38.4|52.0|45.2|
> > > |Delta-CoMe|43.1|51.4|47.3|
> > > |PrinMix|44.1|51.8|48.0|
> > >
> > > - **New evidence on Encoder-Decoder architectures.** We also conducted additional experiments on an Encoder-Decoder model: t5-v1_1-xl (base) and flan-t5-xl (aligned). PrinMix continues to outperform all the baselines.
> > >
> > > |            | MMLU | IFEval | Avg  |
> > > | ---------- | ---- | ------ | ---- |
> > > | Aligned    | 48.7 | 17.0   | 32.9 |
> > > | Low-Rank   | 47.8 | 15.9   | 31.9 |
> > > | BitDelta   | 48.7 | 17.0   | 32.9 |
> > > | Delta-CoMe | 48.3 | 16.1   | 32.2 |
> > > | PrinMix    | 48.8 | 17.7   | 33.3 |
> > >
> > > Together with results on dense decoder-only LLMs and VLMs in the paper, these results provide empirical support that the method generalizes across architectures. We will include these MoE/Encoder-Decoder results in the revision.
> > >
> > > We hope this addresses your concern and would allow you to reconsider the Rebuttal Acknowledgement. Thank you for your valuable input and support.

---

### Official Review · Reviewer_9KYc · 2026-03-12

**Soundness:** 3
**Presentation:** 3
**Significance:** 3
**Originality:** 3
**Overall Recommendation:** 4
**Confidence:** 3

**Summary:**

This paper introduces PRINMIX, an SVD-based mixed-precision delta compression framework for fine-tuned LLMs that frames quantization as an optimization problem. Its key insight is that quantization error in the V matrix depends on singular values, which supports assigning different bit widths to different singular vectors. Building on this, the paper formulates bit allocation as a 0/1 integer linear programming problem under a fixed bit budget and a limit on the number of bit widths used. It also proposes RTC to reduce the errors introduced by sequential quantization.

**Compliance With Llm Reviewing Policy:**

Affirmed.

**Final Justification:**

After considering the paper’s significant strengths, the authors’ comprehensive and compelling rebuttal, and only minor addressable weaknesses, the final recommendation is upgraded from Weak Reject (3) to Weak Accept (4). This adjustment reflects the authors’ effective resolution of all core concerns raised in the initial review.
The paper makes valuable contributions: it proposes PrinMix, a principled SVD-based mixed-precision delta compression framework that models quantization as a mathematical optimization problem, avoids heuristic assumptions, and introduces the novel RTC method to mitigate quantization errors. The authors’ rebuttal fully addressed all major initial concerns, including clarifying the scope of theoretical claims, demonstrating scalability to large models (70B), supplementing critical baselines (DirectC comparison), and decomposing performance advantages.

Only minor revisions are needed: clarifying the scope of the mixed-precision theoretical claim in the main text, integrating supplementary experimental results into the main text, and reinforcing the rationale for U matrix quantization design. These revisions will enhance clarity and rigor without compromising core contributions.

The authors’ rebuttal demonstrates a deep understanding of their work and effectively strengthens the paper’s credibility. The paper’s strengths clearly outweigh its minor weaknesses, and PrinMix provides a practical, principled solution for delta compression of fine-tuned LLMs. We therefore recommend Weak Accept (4), supporting publication after minor revisions.

**Key Questions For Authors:**

1. Can the authors better justify the strength of the theoretical claim around mixed-precision necessity? A careful clarification would improve the paper and avoid overclaiming.
2. How does PRINMIX scale beyond the reported model sizes? Please provide measurements for larger models (e.g., 32B/70B scale), especially the dependence of solve time on layer shape, rank, and candidate bit-width set size.
3. I would like a cleaner decomposition of where the final advantage comes from.
4. How does PRINMIX compare against directly quantizing the aligned model itself (e.g., with GPTQ/AWQ) under matched storage or memory budgets?

**Limitations:**

yes

**Strengths And Weaknesses:**

# Strengths
1. The core contribution starts from a reconstruction-error objective, derives a row-wise error form in the SVD space, and connects that derivation to the mixed-precision design, rather than heuristic singular-value-based allocation rule.
2. RTC is a clean way to mitigate the target mismatch introduced by quantization discarding the GPTQ-style framework.

# Weaknesses
1. The paper overstates the derivation as theoretical proof of the necessity of mixed precision. What it actually shows is something narrower, under its specific setup, the row-wise error for V contains a scaling term that varies with singular values. That is not the same as proving that mixed precision is fundamentally necessary for the true downstream objective.
2. The novelty is meaningful but still somewhat incremental. Relative to Delta-CoMe, the main advance is that the bit allocation is optimized rather than heuristic.
3. The handling of U is still partly heuristic. Section 3.1.2 argues mixed precision is unnecessary for U and then applies the same schedule from V.  This design choice lacks rigor, and it remains unproven whether optimizing the error for V alone is a near-optimal strategy for reducing overall model error.
4. The evaluation misses a highly practical baseline: directly quantizing the fully aligned model(like GPTQ or AWQ). Without comparing the proposed SVD+ILP pipeline to these standard methods under the same memory constraints, it is difficult for practitioners to know if the added complexity of this delta-compression approach is actually worth it.

---

> ### Author Rebuttal · Authors · 2026-03-31
>
> > ### W1&Q1:Scope of theoretical proof on the necessity of mixed precision
>
> R1:We apologize for any confusion caused by our wording. To clarify, we did not intend to claim "mixed precision is fundamentally necessary for the true downstream objective"; Our proof of the necessity of mixed precision is limited to minimizing layer-wise GPTQ-styled reconstruction error under the SVD decomposition with global bit-budget constraint. We will revise the paper to clarify this point more explicitly
>
> > ### W2:The main advance over Delta-CoMe is that the bit allocation is optimized rather than heuristic
>
> R2:We agree that at the implementation level PrinMix and Delta-CoMe differ in bit-allocation (as well as RTC). However, the core contribution of PrinMix is not “optimized vs. heuristic bit allocation,” but a different and principled motivation for mixed-precision
>
> - **Principled necessity (core contribution)** PrinMix provides a theoretical decomposition of the layer-wise GPTQ-style reconstruction error in the SVD space, which reveals why mixed precision is necessary under this objective. The resulting mixed-precision strategy and RTC are direct consequences of the derivation, rather than an ad hoc design choice.
>
> - **No reliance on the “large singular values are more important” heuristic** Delta-CoMe’s mixed-precision is motivated by the above assumption, which is known to be violated in some cases (lines 89-92). PrinMix does not depend on this assumption
>
> - **Broader practical scope** PrinMix naturally supports multiple compression ratios and compressing LoRA modules, whereas Delta-CoMe cannot.
>
> > ### W3:The handling of U is still partly heuristic
>
> R3:We clarify the rationale for our design
> - **U is insensitive to quantization schedule** Beyong the theoretical evidence in Sec 3.1.2, Table 8 (App C.1) empirically supports this: using single-precision for U vs row/column-wise mixed-precision for U yields similar results
>
> - **Reusing V’s schedule for U is for simplicity** Given (i) U’s demonstrated insensitivity, (ii) U’s columns and V’s rows correspond to the same singular values, we reuse the V's schedule to keep the pipeline simple and avoid extra hyperparameters
>
> > ### W4&Q4:Directly quantizing the fully aligned model
>
> R4:We compare directly quantizing the fully aligned model (DirectC) and PrinMix (quantized base model + 1-bit delta) on Qwen2.5‑Math‑Instruct‑7B. All methods use GPTQ for fairness. N is the number of concurrently deployed aligned models.
>
> |Method|Aligned Bit| Base Bit|Delta Bit|Avg Score|Memory Cost|
> |-|-|-|-|-|-|
> |DirectC|4|-|-|86.5|$4N$|
> |DirectC|2|-|-|2.9|$2N$|
> |PrinMix|-|4|1|84.9|$4+N$|
> |PrinMix|-|8|1|85.2|$8+N$|
>
> - **Memory efficiency at modest scale** Compared to DirectC, PrinMix becomes more memory-efficient once $N \gtrsim 2/3$ (with a 4/8-bit base)
>
> - **Lower I/O cost** In cases offloading idle models to save GPU memory, PrinMix lowers the loading/offloading cost
>
> > ### Q2:Scalability of PrinMix
>
> R5:We report PrinMix on DeepSeek‑R1‑Distill‑70B, using a single L40 GPU. The full quantization $\approx$ 8.4 hours. This one-time cost is acceptable relative to the cost of finetuning a 70B model, and it requires only a single GPU. In addition, the runtime can be reduced by $\approx$ 2x via (i) faster ILP solving and constraining the solution space (Lines 772–783) or (ii) replacing ILP with a dynamic programming solver (R4 to Reviewer vbgL)
>
> We break down the per-transformer-block time (s) for DeepSeek‑R1‑Distill‑7B/14B/70B:
>
> |Model|Hidden Size|Intermediate Size|QK Size|Simulation|Optimization|Quantization|Total Time|
> |-|-|-|-|-|-|-|-|
> |7B|3584|18944|512|31.2|72.5|20.7|124.4|
> |14B|5120|13824|1024|33.4|128.6|26.0|188.0|
> |70B|8192|28672|1024|141.0|175.5|59.9|376.4
>
> * **Rank sensitivity**
>
> |size|Rank|Simulation|Optimization|Quantization|Total|
> |-|-|-|-|-|-|
> |14B|128|27.4|14.4|13.5|55.3|
> |14B|256|27.8|14.4|13.6|55.8|
> |7B|128|26.8|13.0|12.1|51.9|
> |7B|256|26.3|13.8|14.3|54.4|
>
> * **Candidate bit-width set size**
>
> |#Candidate Bit-widths|Simulation|Optimization|Quantization|Total|
> |--|-|-|-|-|
> |4|31.0|35.7|20.2|86.9|
> |6|31.5|50.4|20.4|102.3|
> |8|31.2|72.5|20.7|124.4|
>
> These results show that (i) solve time is weakly sensitive to rank, (ii) it increases with the layer size and candidate bit-width set size
>
> > ### Q3: Cleaner decomposition of where the final advantage comes from
>
> R6:
> We provide a decomposition of the gains at two levels:
> - **Component ablations (Table 5)**  At implementation-level, PrinMix is different from Delta‑CoMe with bit-allocation and RTC. Table 5 isolates these components and shows that (i) our bit allocation contributes +1.2 and (ii) RTC yields a further +1.3 improvement
>
> - **Attribution within bit allocation (U vs. V) (Table 8, App. C.1)** We further disentangle where the mixed-precision gain comes from by applying different quantization strategies to U. Table 8 shows that different strategies perform similarly on U, indicating that improvement is primarily from mixed-precision of V

---

> > ### Author Rebuttal · Reviewer_9KYc · 2026-04-03
> >
> > Thank you for your rebuttal. My confusion regarding the experimental section has been resolved. I will raise score.

---

> > > ### Author Response · Authors · 2026-04-04
> > >
> > > Thank you for your positive feedback. We are glad that our responses have addressed your concerns, and we greatly value your recognition and suggestions, which helps us to further improve our paper. We will make sure to incorporate them into the revised version.
> > >
> > > We also note that the updated score does not yet appear in the system. We would be grateful if you could kindly confirm this, as it matters a great deal to us. If you have any further questions, we would be more than happy to provide additional information. Thank you again for your time and support.

---

### Decision · Program_Chairs · 2026-04-30

**Decision:**

Accept (regular)

**Comment:**

This paper proposes techniques to improve delta compression, which decomposes LLMs into base and delta weights, with the delta compressed using SVD-based mixed-precision quantization. Specifically, the paper formulates bit allocation as a 0/1 integer linear programming problem and further proposed a Reconstruction Target Correction (RTC) method to compensate for errors introduce by the sequential V-then-U quantization process. The framework is evaluated on popular LLM model families across multiple tasks, demonstrating improved performance over previous delta-compression techniques.

The paper is reasonably well written, and all reviewers give favorable ratings. The main strength lies in its detailed theoretical analysis of quantization error in the SVD space, which provides a solid theoretical justification for the mix-precision design and motivates the RTC method to avoid bias introduced by sequential V-then-U quantization. In addition, the evaluation is reasonably thorough. The experimental results show clear improvement over prior techniques while avoiding heuristic assumptions.

During rebuttal, the authors provide additional clarification on the scope of the theoretical claim, the heuristic handling of U, and numerical stability of the RTC method, along with additional results across more model architectures and scales. These addressed most of the concerns raised by the reviewers. Therefore, this paper is recommended acceptance.